# Cardiolipin exposure on the outer mitochondrial membrane modulates α-synuclein

Tammy Ryan[1], Vladimir V. Bamm[1], Morgan G. Stykel[1], Carla L. Coackley[1], Kayla M. Humphries[1], Rhiannon Jamieson-Williams[1], Rajesh Ambasudhan[2], Dick D. Mosser[1], Stuart A. Lipton[2,3,4], George Harauz [1] & Scott D. Ryan[1,2]

Neuronal loss in Parkinson's disease (PD) is associated with aberrant mitochondrial function and impaired proteostasis. Identifying the mechanisms that link these pathologies is critical to furthering our understanding of PD pathogenesis. Using human pluripotent stem cells (hPSCs) that allow comparison of cells expressing mutant *SNCA* (encoding α-synuclein (α-syn)) with isogenic controls, or *SNCA*-transgenic mice, we show that *SNCA*-mutant neurons display fragmented mitochondria and accumulate α-syn deposits that cluster to mitochondrial membranes in response to exposure of cardiolipin on the mitochondrial surface. Whereas exposed cardiolipin specifically binds to and facilitates refolding of α-syn fibrils, prolonged cardiolipin exposure in *SNCA*-mutants initiates recruitment of LC3 to the mitochondria and mitophagy. Moreover, we find that co-culture of *SNCA*-mutant neurons with their isogenic controls results in transmission of α-syn pathology coincident with mitochondrial pathology in control neurons. Transmission of pathology is effectively blocked using an anti-α-syn monoclonal antibody (mAb), consistent with cell-to-cell seeding of α-syn.

[1] Department of Molecular and Cellular Biology, The University of Guelph, Guelph, ON N1G 2W1, Canada. [2] Neurodegenerative Disease Center, Scintillon Institute, 6868 Nancy Ridge Drive, San Diego, CA 92121, USA. [3] Department of Neurosciences, University of California, San Diego School of Medicine, La Jolla, CA 92093, USA. [4] Departments of Molecular Medicine and Neuroscience, and Neuroscience Translational Center, The Scripps Research Institute, La Jolla, CA 92037, USA. Tammy Ryan and Vladimir V. Bamm contributed equally to this work. Correspondence and requests for materials should be addressed to S.D.R. (email: sryan03@uoguelph.ca)

The decline of voluntary motor function in Parkinson's disease (PD) is caused by loss of A9 dopaminergic (DA) neurons in the substantia nigra pars compacta. The death of these neurons is preceded by the formation of intracellular proteinaceous aggregates known as Lewy bodies, which contain a variety of misfolded protein components, including α-synuclein (α-syn)[1]. Although aggregated proteins were first considered to be pathogenic, evidence suggests that macroscopic aggregates are an attempt by the cell to sequester aberrant proteins, whereas soluble oligomers (micro-aggregates) of such proteins are the most toxic forms (for a review see ref. [2]).

The physiological conformation of α-syn remains contentious. Recently, in-cell nuclear magnetic resonance studies have shown that α-syn in neuronal cells is intrinsically disordered[3]. Although this is a prevailing hypothesis, several other studies have suggested that α-syn exists in a tetrameric form within the cytosol of human brain, whereas membrane-associated α-syn is monomeric[4,5]. The interaction of α-syn with negatively charged phospholipids in membranes promotes the adoption of an α-helical structure[6]; however, α-syn can also form oligomers of pleated β-sheets which can subsequently fibrilize and contribute to Lewy bodies[7,8]. Interestingly, the A53T and E46K mutations in α-syn, which have been associated with early-onset familial PD, accelerate β-sheet formation and fibrilization[7,9,10]. These findings suggest that the identification of mechanisms to promote normal folding of α-syn may be of therapeutic interest. Indeed, although mutation of the SNCA gene is causal in several rare familial forms of PD, mutations in the SNCA locus are also consistently identified in genome-wide association studies of sporadic PD[11], and α-syn protein is commonly found in Lewy neurites of idiopathic disease cases. These findings emphasize the need to understand how altered α-syn structure and function relate to PD pathogenesis in general terms.

In addition to protein misfolding and impaired proteostasis, transgenic animal studies of PD-associated gene mutations that focused on the molecular basis of neuronal loss found PD pathology to be a consequence of mitochondrial damage and oxidative stress[1,12]. In many neurodegenerative diseases, mitochondria become fragmented, which is ultimately believed to promote their clearance by the cell's autophagic machinery via a process termed mitophagy[13]; however, excessive or prolonged activation of mitophagy may, in and of itself, contribute to neurodegeneration. Although data supporting mitochondrial dysfunction in PD are abundant, the link between these defects and synucleinopathy remains unclear. α-Syn has been reported to bind to both mitochondrial membranes[14] and mitochondria-associated endoplasmic reticulum (ER) membranes[15], where it contributes to mitochondrial fragmentation. The association of α-syn between these membranes, however, is altered by SNCA mutation[15], and the importance of membrane binding to α-syn function remains obscure.

The generation of human isogenic induced pluripotent stem cell (hiPSC) and embryonic stem cell (hESC) models of familial PD has facilitated the analysis of PD pathology at the cellular level[16], allowing us to contrast A9-type DA neurons (hNs) harboring the SNCA-A53T mutation with isogenic, mutation-corrected controls[12]. Using these systems, we describe a novel mechanism in which the mitochondrial membrane lipid cardiolipin plays a vital role in the folding of α-syn protein. We demonstrate that cardiolipin translocates to the outer mitochondrial membrane (OMM) in both A53T and E46K SNCA-mutant hNs, where it binds mutant α-syn, facilitating the folding of α-syn to an α-helix. This finding was confirmed in SNCA-A53T transgenic mouse brains. Furthermore, we show that OMM-localized cardiolipin can pull α-syn monomers away from oligomeric fibrils and facilitate their refolding from aggregated

β-sheet forms back to monomers comprising α-helices, effectively buffering synucleinopathy. Surprisingly, both the A53T and E46K α-syn mutations reduce the kinetic rate of cardiolipin-mediated α-syn refolding relative to wild type (WT), increasing the level of α-syn present at the OMM in A53T and E46K hNs. Binding of A53T and E46K α-syn to cardiolipin leads to significantly increased LC3 recruitment to the OMM relative to WT α-syn, which in turn increases mitochondrial stress, triggering excessive mitophagy. Finally, we show that mitochondrial pathology is transmissible from α-syn mutant cells to non-mutant cells by direct seeding of α-syn protein, a process that can be blocked with a monoclonal antibody (mAb) against α-syn. Using hPSC-derived A9-type DA neurons, this work represents the first demonstration of direct cell-to-cell transmission of endogenous, mutant α-syn from diseased human neurons to non-diseased human neurons, and identifies cardiolipin exposure on mitochondrial membranes as a key signal in PD pathogenesis.

## Results

**Neurons from SNCA-mutant hPSCs demonstrate synucleinopathy.** In order to evaluate the relationship between α-syn folding and mitochondrial integrity in PD, we required a homogeneous population of A9-type DA neurons. Using either hiPSC-derived clones that allow for comparison of the A53T-SNCA mutation (A53T) with isogenic-corrected controls (Corr)[12,16] or hESC-derived clones that allow for comparison of WT-SNCA with A53T-SNCA or E46K-SNCA mutations, we characterized the lineage progression of hPSCs to DA neurons. Employing a floor-plate induction protocol[17], we first differentiated A53T and corrected hiPSCs into human A9-type DA neurons that were both microtubule-associated protein 2/tyrosine hydroxylase (TH) double positive and G-protein-regulated inward-rectifier potassium channel 2 (Girk2)/TH double positive (Supplementary Fig. 1a). Quantification of these cell types within our cultures showed that our differentiation was robust, yielding over 80% TH +ve DA neurons, most of which were also positive for Girk2 (Supplementary Fig. 1b). Importantly, and consistent with previous reports, there was no difference in the proportion of DA neurons between corrected and A53T cultures (Supplementary Fig. 1b)[18]. Labeling of corrected and A53T cultures with antibodies against vGlut/Tuj1 and GAD65/Tuj1 revealed that a much smaller proportion of our hN cultures expressed these markers relative to that expressing TH/Tuj1 (Supplementary Fig. 1c–f). We next sought to characterize the level of synucleinopathy exhibited by SNCA-mutant (A53T and E46K) hNs. To determine the physical state of proteins deposits, we performed a TX-100 wash-out of soluble protein in both hiPSC (Fig. 1a) and hESC (Fig. 1b)-derived hNs prior to fixation. We labeled these hNs with antibodies against ubiquitin, total α-syn and α-syn phosphoserine 129 (α-syn PS129), a modification commonly associated with synucleinopathy[19,20]. We found that the accumulations of α-syn PS129 and ubiquitin in SNCA-mutant lines were in TX-100 insoluble protein deposits (Fig. 1a, b). We further characterized the proximity of α-syn PS129 with ubiquitin modifications by fluorescence resonance energy transfer (FRET) and found a significant increase in the level of energy transfer in between 488-labeled ubiquitin and 594-labeled PS129 and in SNCA-mutant hNs relative to their respective controls (Fig. 1c, d), indicating that these modifications reside in very close proximity. Taken together, these results indicate that both hiPSC and hESC-derived hNs harboring SNCA mutations exhibit early hallmarks of synucleinopathy.

**SNCA-mutant hNs have impaired mitochondrial dynamics.** Having established that SNCA-mutant hNs exhibit early evidence

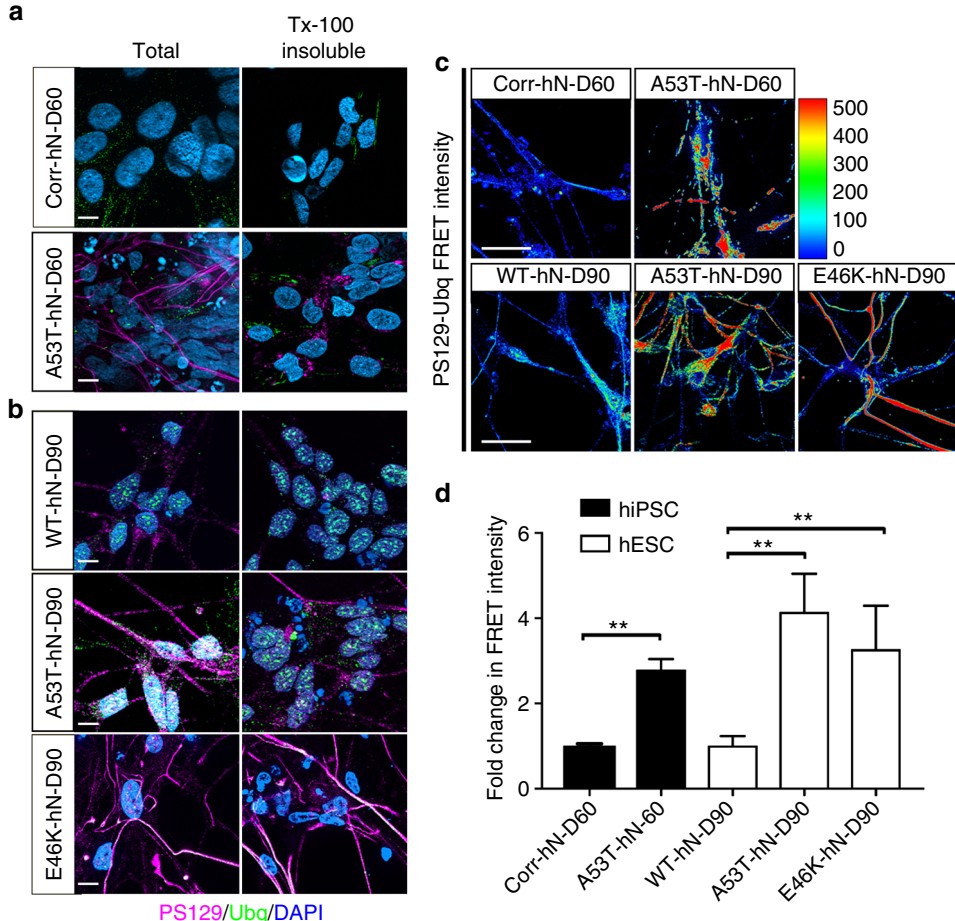

**Fig. 1** α-Syn mutant hNs acquire early signs of α-syn pathology. **a**, **b** Accumulation of insoluble Ubq/α-syn PS129 protein in hiPSC-derived A53T and corrected hNs (**a**) or hESC-derived WT, A53T and E46K hNs (**b**) was determined by TX-100 wash-out of soluble protein prior to fixation. DIV: 60. Scale bar: 10 µm. **c**, **d** Fluorescence resonance energy transfer (FRET) from Alexa-488-labeled ubiquitin to Alexa-594-labeled α-syn PS129 in hiPSC-derived A53T and corrected hNs or hESC-derived WT, A53T and E46K hNs was assessed and mean FRET intensity was quantified (**d**). Data represent mean ± s.e.m. \*\**P* < 0.01 by ANOVA followed by Tukey's post hoc test, *n* = 6 coverslips over three independent differentiations, DIV: 60 or 90. Scale bar: 50 µm

of synucleinopathy, we next sought to examine mitochondrial function and dynamics in this system. We first looked at the structural integrity of mitochondria in hiPSC-derived corrected and A53T hNs (Fig. 2a, c) as well as in hESC-derived WT, A53T and E46K hNs (Fig. 2b, c) expressing mitochondrial-targeted DSRed (mitoDSRed) using structured illumination-based microscopy[21]. While control hNs exhibited uniform ribbon-like mitochondrial morphology characteristic of healthy neurons, *SNCA*-mutant hNs from both hiPSC and hESC backgrounds contained highly fragmented mitochondria (Fig. 2c) consistent with our previous observations of impaired mitochondrial bioenergetics in A53T hNs[12]. Next, we assessed mitochondrial morphology by transmission electron microscopy (TEM), and we found that mitochondria in A53T hNs had a significantly reduced diameter when compared to isogenically corrected controls (Fig. 2d, f) as did A53T and E46K hNs relative to WT hNs (Fig. 2e, f).

Aberrant interactions with mutated or misfolded α-syn are thought to contribute to mitochondrial dysfunction in PD[14,22]. We therefore investigated the localization of PS129-modified α-syn in A53T and corrected hNs with respect to mitochondria. The hNs expressing mitoDSRed were antigenically labeled for α-syn PS129, and signal colocalization was measured by optical sectioning. While corrected hNs had very little PS129 labeling and displayed elongated mitochondrial morphology, A53T cultures contained a significantly higher percentage of neurons

with α-syn PS129 in close proximity to fragmented mitochondrial membranes, suggesting that mutant α-syn is closely associated with dysfunctional mitochondria (Fig. 2g). Quantitatively, A53T cultures had far more cells with α-syn PS129 on fragmented mitochondria than did corrected hNs (Fig. 2h). We confirmed the proximity of α-syn to the OMM using the proximity ligation assay. α-Syn molecules within 40 nm of the OMM marker Tom40 fluoresce red when excited with a 543 nm laser. Using this approach, we found extensive interaction between OMM-localized Tom40 and α-syn in A53T hNs but only modest signal in corrected controls (Fig. 2i, j), suggesting that α-syn is closely associated with the fragmented mitochondrial membranes in these cells. By contrast, no enrichment of PS129-modified α-syn could be identified on ER or Golgi membranes (Supplementary Fig 2a, b).

Our ultrastructural analysis of mitochondria by TEM frequently showed mitochondria from *SNCA*-mutant hNs in close proximity with vacuoles (Fig. 2d, e), which could indicate increased mitophagy. In order to further characterize mitochondrial integrity in this system, we assessed the formation of LC3[+ve] puncta, which when colocalized with mitochondria, indicate targeting of mitochondria to the autophagic machinery. We antigenically labeled mitoDSRed-expressing hNs for LC3, and found that while genome corrected control hNs exhibited a low level of diffuse LC3 labeling, A53T hNs showed extensive LC3[+ve]

puncta that colocalized with mitoDSRed (Fig. 3a, b). LC3 conversion and induction were also tracked (Supplementary Fig 3a, b), using starvation to induce conversion of LC3I to LC3II and the lysosomal inhibitor chloroquine to measure steady-state levels of LC3II. A53T neurons showed a higher rate of LC3 turnover and lower threshold for LC3 conversion than isogenic controls. We next assessed the functional response of mitochondria to the uncoupling agent carbonyl cyanide-4-(trifluoromethoxy)phenylhydrazone (FCCP) in A53T and corrected hNs. At baseline, A53T hNs had a higher number of LC3$^{+ve}$ puncta per cell compared to corrected hNs (Fig. 3c). Exposure of A53T hNs to FCCP over a 360-min time-course resulted in a more

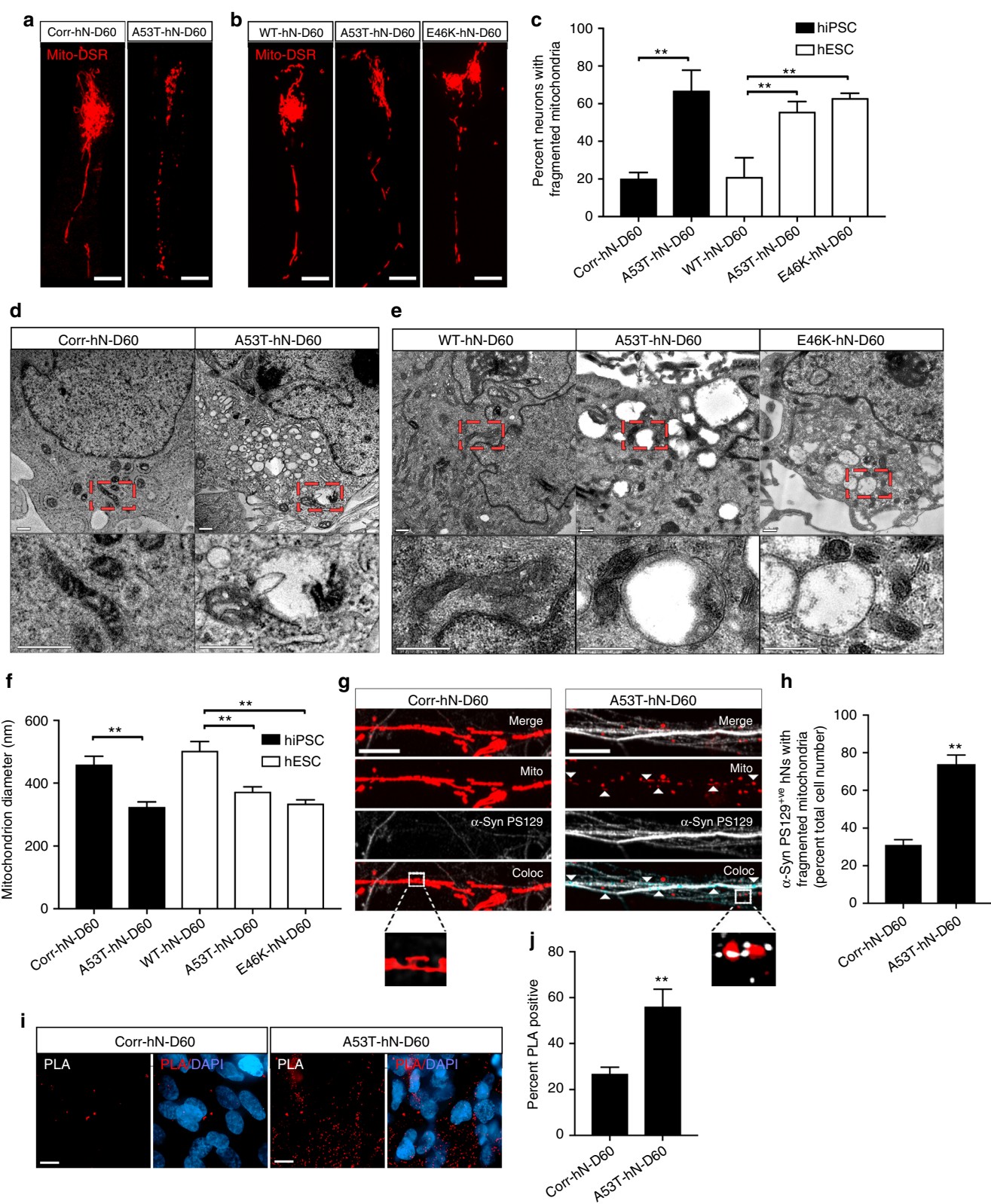

rapid initial increase in the number of LC3$^{+ve}$ puncta than that observed in corrected cells (Fig. 3c), but ultimately led to complete clearance of MitoDSRed-labeled mitochondria in both lines (Fig. 3d).

Next, as an indicator of mitochondrial function and stress, we measured mitochondrial potential in A53T and corrected hNs by staining with tetramethylrhodamine ethyl ester (TMRE) via the dequench method[23], in which loss of mitochondrial potential leads to dequenching of TMRE fluorescence at the mitochondria and a corresponding increase in cytoplasmic fluorescence. Corrected and A53T hNs were labeled with TMRE and analyzed by flow cytometry (Fig. 3e). The A53T hNs showed increased levels of TMRE fluorescence relative to corrected controls, indicating that the mitochondria in these cells have reduced membrane potential (Fig. 3e, f). Taken together, these data indicate that both the structure and function of mitochondria in PD patient-derived A53T hNs are severely compromised relative to corrected controls.

Finally, we assessed mitophagic flux in each of our *SNCA*-mutant lines (both hiPSC and hESC derived) relative to their respective isogenic controls using the mt-mKeima assay to measure mitochondria targeted to the acidic compartment of the autolysosome (Fig. 3g, h). In this assay, targeting of mKeima to the lysosomal compartment results in a shift in fluorescence excitation from 438 nm (red) to 561 nm (white). A53T and E46K hNs from both hiPSC and hESC origins had a higher ratio of 561 nm to 438 nm fluorescence than their isogenic counterparts (Fig. 3g, h), supporting an overall increase in the rate of mitophagy in *SNCA*-mutant hNs. Furthermore, knockdown of the mitophagic modulator Beclin-1, resulted in rescue of mitochondrial volume in A53T hNs, further implicating mitophagy in α-syn-induced mitochondrial dysfunction (Supplementary Fig. 3c, d).

**Cardiolipin is externalized to the OMM in *SNCA*-mutant hNs.** At the developmental time point of analysis of 60 days in vitro (DIV), mitochondrial pathology/fragmentation was already significant in A53T hNs, and considerable α-syn deposits had accumulated at mitochondrial membranes. To determine whether α-syn was interacting with mitochondria prior to onset of mitochondrial fragmentation and triggering dysfunction, we took advantage of the differentiation potential of our hPSC systems, and examined cells prior to deposition of synucleinopathy. We began by measuring the relative level of expression of α-syn in A53T and Corr cells at the nestin-positive precursor stage (DIV 14). We found that α-syn levels were higher in A53T cells than corrected cells (Fig. 4a), consistent with previous reports[12], and that this accumulation had begun by DIV 14 and persisted through differentiation in hiPSC (Fig. 4a) and hESC-derived (Supplementary Fig. 4a) *SNCA*-mutant hNs. This observation is

consistent with recent biochemical studies that have revealed that accumulation of monomeric α-syn was a predictor of disease pathology in α-syn transgenic mouse brain and hiPSC-derived A53T hNs[5].

In order to understand the relevance of α-syn accumulation with respect to mitochondrial pathology, we next assessed biochemically the accumulation of the various structural forms of α-syn on isolated mitochondria over the course of neuronal differentiation. We used TX-100 to separate soluble mitochondrial proteins from TX-100 insoluble, unfolded protein aggregates, which can be subsequently solubilized in urea for analysis (Fig. 4b). Western blot analysis of mitochondria from DIV 60 corrected and A53T hNs showed an increased accumulation of TX-100 soluble α-syn in A53T hNs relative to corrected hNs (Fig. 4b, c). Longer exposure showed that this was preceded by the accumulation of soluble α-syn in the mitochondrial fractions of DIV 14 A53T NPCs relative to corrected NPCs (Fig. 4b, c). Interestingly, no insoluble deposits were observed in mitochondrial fractions. This result suggests that PD-causing mutations may promote the accumulation of soluble synuclein at the mitochondria prior to deposition of insoluble aggregates elsewhere in the cell.

In functional mitochondria, the negatively charged phospholipid cardiolipin (CL) localizes to the inner mitochondrial membrane (IMM) but translocates to the OMM in response to cellular stress, providing an anchor for LC3 and initiating mitophagy[24,25]. Given the association of α-syn with fragmented mitochondria (Fig. 2g), the early accumulation of monomeric α-syn in mitochondrial fractions of A53T hNs (Fig. 4a–c) and previous studies showing that α-syn interacts with cardiolipin at the mitochondria[14], we hypothesized that cardiolipin externalization in response to the presence of α-syn may mark an early step in the events leading to impaired mitochondrial dynamics in A53T hNs.

We therefore sought to assess cardiolipin localization within both *SNCA*-mutant cells and *SNCA*-transgenic animals. Cardiolipin is an anionic phospholipid whose externalization has been shown to dramatically lower the surface charge of the mitochondria[26]. To assess cardiolipin externalization, we first used a probe for anionic charge (RPRE-RFP)[26,27]. Under physiological conditions, the most abundant anionic charge in a cell is phosphatidylserine at the inner surface of the plasma membrane. Upon cardiolipin externalization, RPRE-RFP redistributes from the plasma membrane to intracellular sites, primarily mitochondria[26,27]. In order to assess whether the mitochondria in our system were externalizing cardiolipin prior to the onset of mitophagy, we first expressed mitoDSRed in DIV 14 corrected and A53T NPCs and assessed mitochondrial structure (Supplementary Fig. 4b, c). We found no difference in the number of cells with fragmented mitochondria or in the total

**Fig. 2** Defects in mitochondrial structure and function in α-syn mutant hNs. **a, b** hiPSC-derived A53T hNs (**a**) and hESC-derived A53T and E46K hNs (**b**) have more highly fragmented mitochondria compared to corrected hNs and WT hNs respectively as shown by mitoDSRed expression; scale bar: 10 μm. **c** Percentage of total hNs that have fragmented mitochondria. Data represent mean ± s.e.m. **P < 0.01 by ANOVA followed by Tukey's post hoc test, for hiPSCs, *n* = 12 coverslips, for hESCs, *n* = 6 coverslips from 3 independent differentiations, DIV: 60. **d-f** Transmission electron micrographs of hiPSC-derived corrected and A53T hNs (**d**) or hESC-derived WT, A53T and E46K hNs (**e**) show that mutant cells contain smaller mitochondria, with a lower average diameter (**f**), **P < 0.01 by ANOVA followed by Tukey's post hoc test, *n* = 6 independent cultures from 3 independent differentiations. **g** Micrographs of neurites from corrected (left panels) and A53T-mutant (right panels) hNs expressing mitoDSRed and labeled for α-syn phosphoserine 129 (PS129). Arrows show areas where PS129 colocalizes with MitoDSRed. Boolean operation was employed to pseudo-label regions of colocalization in blue. Boxes show high magnification of the regions indicated. Scale bar: 10 μm. **h** Quantification of hNs with α-syn PS129 on fragmented mitochondria. Data represent mean ± s.e.m. **P < 0.0001 by *t*-test, *n* = 6, DIV: 60. **i, j** Proximity of Tom20 and α-syn as assessed by proximity ligation assay (PLA) (**i**) showed significantly more α-syn within 40 nm of the OMM in A53T relative to corrected hNs (**j**). **P = 0.0055 by Student's *t*-test, *n* = 6 coverslips from 3 independent differentiations, DIV: 60, scale bar: 10 μm

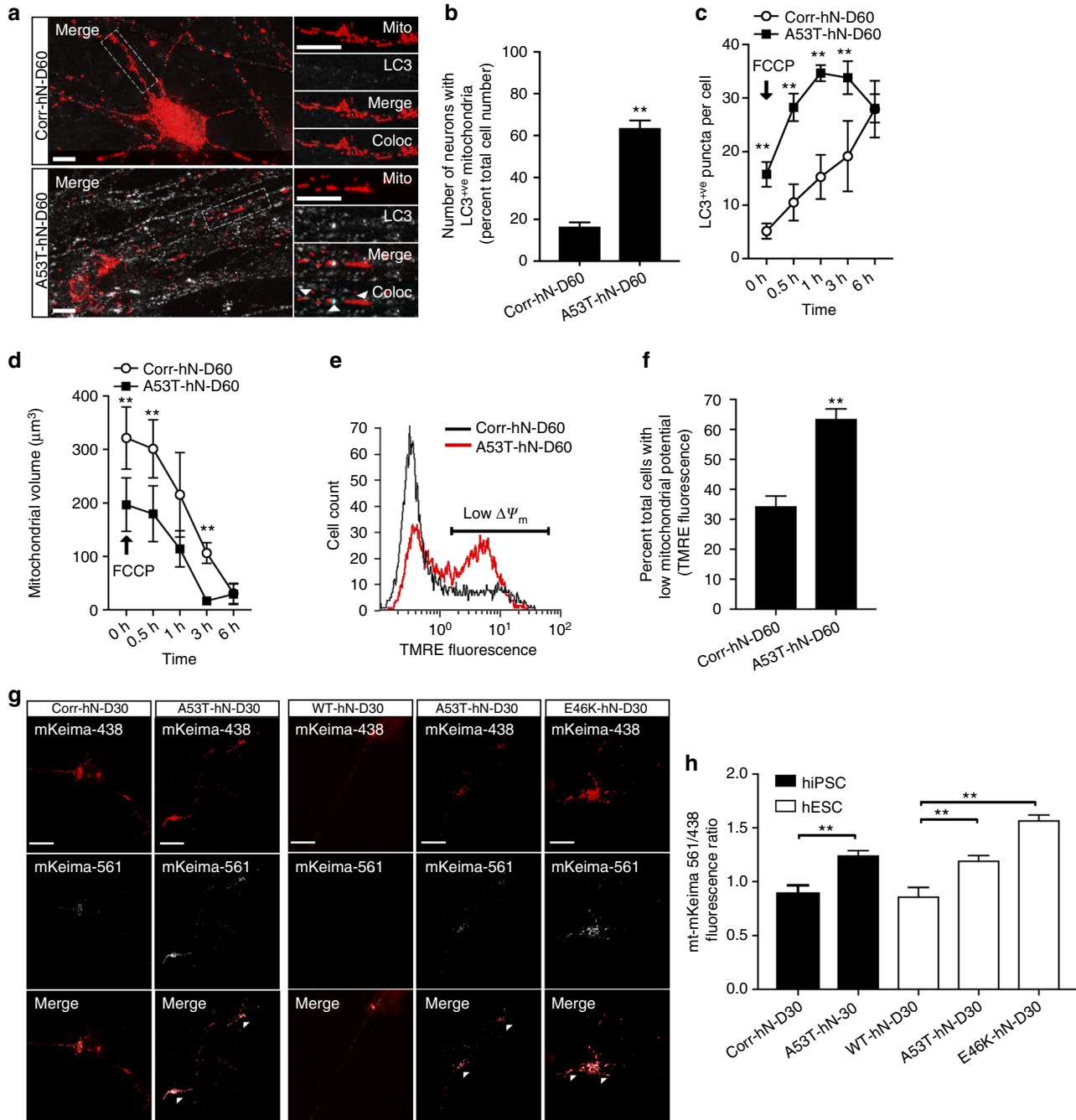

**Fig. 3** α-Syn-mutant hNs show evidence of increased mitophagic turnover. **a**, **b** A53T hNs display more punctate LC3 staining localizing to mitochondria relative to corrected hNs (top panels) (**a**). Arrows show areas where LC3 colocalizes with MitoDSRed. Boolean operation was employed to label regions of colocalization in blue, scale bar: 10 μm. Quantification of cells containing LC3-positive mitochondria (**b**). Data represent mean ± s.e.m. **$P = 0.0002$ by $t$-test, $n = 6$ coverslips from 2 independent differentiations, DIV: 60. **c**, **d** MitoDSRed expressing A53T and corrected hNs were exposed to FCCP and the number of LC3$^{+ve}$ puncta per neuron (**c**) coupled to the total mitochondrial volume (**d**) was measured over time (360 min). LC3$^{+ve}$ puncta form more rapidly in A53T-mutant hNs relative to corrected hNs corresponding to an accelerated loss of mitochondria. Data represent mean ± s.e.m. **$P < 0.01$ by repeated measures ANOVA with multiple t-test, $n = 3$ coverslips from 2 independent differentiations, DIV: 60. **e**, **f** A53T and corrected hNs were labeled with 200 nM TMRE, and mitochondrial potential ($\Delta\Psi_m$) was measured by flow cytometry using the dequench method. Representative plot showing forward scatter and TMRE fluorescence as a measure of cell size and $\Delta\Psi_m$, respectively (**e**), as well as the percent depolarized cells (**f**) illustrate that A53T hNs have lower mitochondrial potential than corrected hNs. **$P = 0.0039$ by $t$-test, $n = 3$ coverslips from 3 independent differentiations, DIV: 60. **g**, **h** Autophagic flux was measured by mt-mKeima targeted to the autolysosome. Targeting of mKeima to the lysosomal compartment results in a florescence shift in excitation from 438 nm (red) to 561 nm (white) (**g**). hiPSC-derived A53T hNs and hESC-derived A53T and E46K hNs had a higher ratio of 561 nm to 438 nm fluorescence than their corrected or WT counterparts, respectively (**h**). **$P < 0.01$ ANOVA followed by Tukey's post hoc test, for hiPSC $n = 12$, for hESCs, $n = 8$ coverslips from 3 independent differentiations, DIV: 60

mitochondrial volume per cell between A53T cells and their genetically corrected controls at this developmental stage. Next, we expressed the RPRE-RFP charge probe in both A53T and control NPCs at DIV 14 and assessed its subcellular localization. While RPRE-RFP was predominantly localized to the plasma membrane in genetically corrected cells, A53T cells showed a marked shift in subcellular localization of RPRE-RFP to intracellular puncta (Supplementary Fig. 4d, e), indicating that intracellular membranes in A53T hNs had begun to accumulate negative charge.

We next sought to determine whether accumulation of intracellular negative charge could be attributed to cardiolipin

exposure on the OMM. We visualized cardiolipin using a green fluorescent protein (GFP)-labeled cardiolipin probe (CL-GFP) that contains the cardiolipin-binding domain of the mitochondrial protein SLP-2, in conjunction with RPRE-RFP to assess anionic charge. We found a dramatic increase in colocalization of the two probes in A53T cells relative to corrected cells (Fig. 4d, e), consistent with the notion that the increase in negative charge at the mitochondrial OMM in A53T cells is due to cardiolipin externalization. We further characterized the proximity of RPRE to cardiolipin by FRET in both hiPSC-derived A53T and corrected cells as well as in hESC-derived WT, A53T and E46K cells. We found a significant increase in the level of energy

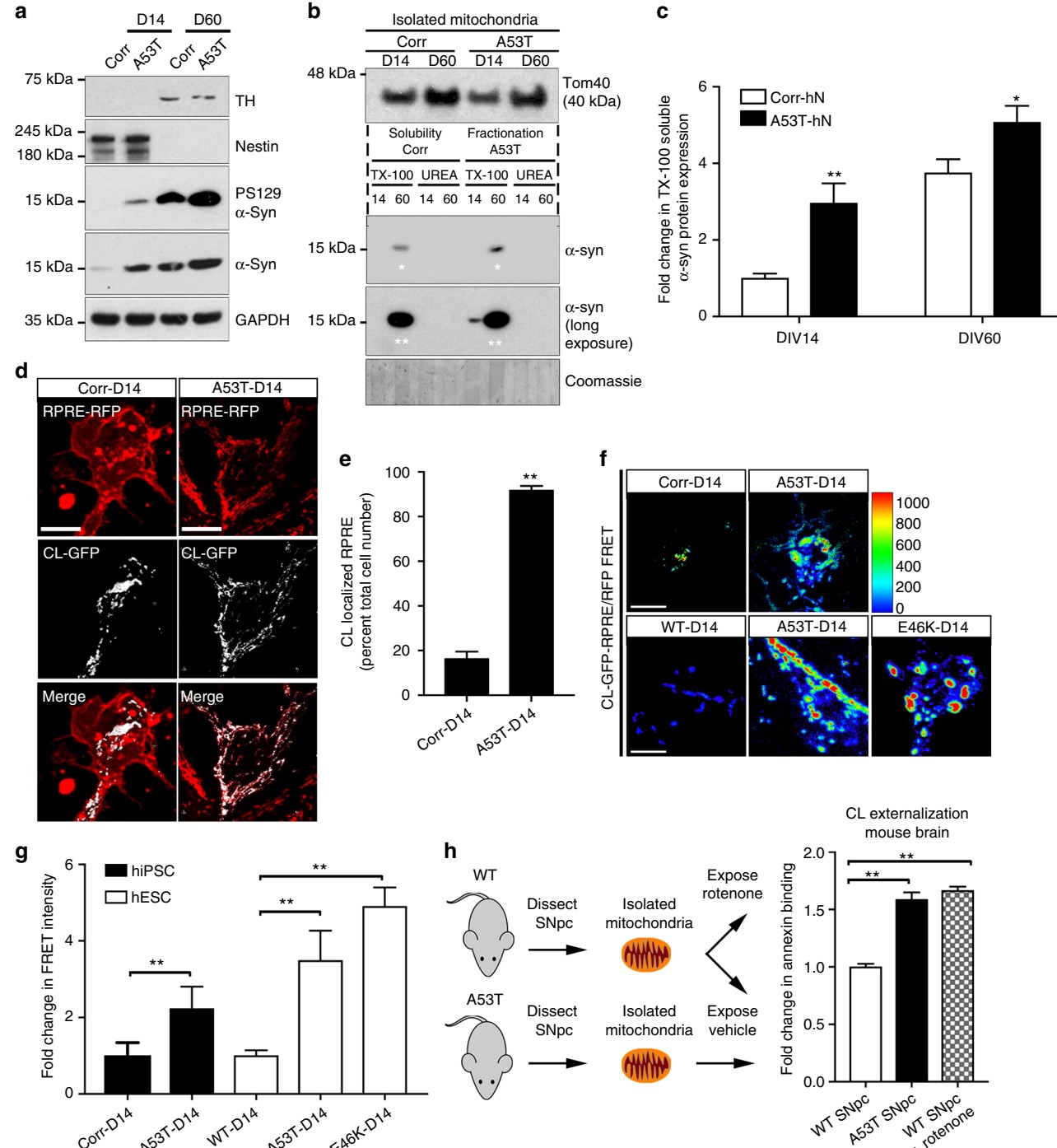

transfer between GFP-labeled cardiolipin and RFP-labeled RPRE in *SNCA*-mutant hNs relative to their respective controls (Fig. 4f, g), indicating that these molecules reside in very close proximity. Finally, we sought to confirm this finding in vivo, using *SNCA*-A53T mice expressing human A53T-mutant α-syn under the Thy1 promoter[12,28,29]. We observed accumulation of heat-stable α-syn in conjunction with α-syn-ubiquitylation in these animals by the age of 6 months (Supplementary Fig. 4f). Mitochondria were isolated from the SNpc of 6-month-old WT and A53T transgenic animals and labeled with Annexin V, which is effective in measuring cardiolipin translocation to the OMM in isolated mitochondria[25,30]. We found that mitochondria from *SNCA*-A53T transgenic animals show significantly more cardiolipin on the mitochondrial surface than their WT littermates. Additionally, we found that *SNCA*-A53T animals externalized cardiolipin on the OMM to a similar extent to that of WT mitochondria that had been exposed to rotenone (Fig. 4h).

Collectively, these data indicate that while the *SNCA*-mutation is associated with mitochondrial stress in mature neurons both in vitro and in vivo, this stress is initiated at earlier developmental stages in response to α-syn present at mitochondrial membranes. This association is coincident with a loss of mitochondrial surface charge that results from cardiolipin exposure on the OMM and precedes any increase in mitochondrial fragmentation. This phenomenon may thus represent an early marker of pathology that ultimately leads to excessive mitophagy due to prolonged cardiolipin exposure in mature A53T hNs.

**Cardiolipin binds α-syn and refolds α-syn oligomers.** Given our findings that cardiolipin exposure occurs well before any mitochondrial pathology, we sought to understand the purpose of cardiolipin externalization in response to α-syn present on mitochondrial membranes. We postulated that cardiolipin was involved in regulating either (1) binding of mitochondrial membranes to α-syn, (2) folding of disordered α-syn into an α-helical state or (3) refolding of oligomeric/fibrillar α-syn aggregates. In a cell-free system, we conducted a series of proof-of-principle experiments in which we generated large unilamellar lipid vesicles (LUVs) with increasing molar ratios of cardiolipin that mimic the composition of the OMM (OMM vesicles). These cardiolipin concentrations (8–30%) centered on the levels of cardiolipin known to be externalized to the OMM of neurons after exposure to the mitochondrial complex I inhibitor rotenone[25]. Using circular dichroic spectroscopy (CD), we first assessed the ability of WT, A53T and E46K human recombinant α-syn to bind cardiolipin-containing LUVs (Fig. 5a–e). Regression analysis of binding curves showed that WT and mutant α-

syn bind OMM LUVs in a cardiolipin-dependent manner, albeit with differing specificities (Supplementary Table 1). Moreover, all three α-syn forms had the same affinity for cardiolipin at a 30% molar ratio, consistent with the level of cardiolipin reported to recruit LC3 upon induction of mitochondrial stress[25,30]. We simultaneously assessed the ability of LUVs containing increasing molar ratios of cardiolipin to fold intrinsically disordered α-syn monomers (Supplementary Fig. 5). We found a shift in the spectra of both WT and mutant α-syn from disordered random coil to α-helix with increasing cardiolipin ratios, indicating that cardiolipin in the OMM can both bind to and promote folding of disordered monomeric α-syn.

To address whether this mechanism occurs in the context of pathology, we synthesized artificial, preformed fibrils (PFFs) from α-syn monomers and conducted a similar experiment to determine if cardiolipin externalization can facilitate refolding of α-syn fibrils. We incubated WT, A53T or E46K preformed α-syn fibrils with LUVs containing 30% cardiolipin by molar ratio (Fig. 5f–h). Time-dependent analysis by CD spectroscopy revealed that cardiolipin-containing LUVs are indeed able to pull WT, A53T and E46K α-syn monomers away from fibrils and refold the protein. These data were compared against the CD spectra of monomeric protein established in Fig. 5e. Kinetic analysis of the data at 222 nm (a wavelength informative of α-helical formation) showed that the rate of refolding in the presence of cardiolipin was higher for WT than for either A53T or E46K α-syn (Fig. 5i), namely, $3.79\,h^{-1}$ vs. $2.44\,h^{-1}$ vs. $1.30\,h^{-1}$ for WT, A53T and E46K α-syn, respectively. Moreover, deconvolution of the CD spectra at different time points showed that the rate of removal of β-sheet secondary structure components (protein fibrils) from WT, A53T and E46K preformed fibrils was also significantly higher for WT α-syn than for either A53T or E46K α-syn (Fig. 5j), plateauing by 24 h of incubation. Collectively, these data show that α-syn binds OMM vesicles in a cardiolipin-dependent manner and that although cardiolipin-containing OMM vesicles promote refolding of α-syn oligomers, cardiolipin must be externalized for a longer period of time to refold A53T and E46K α-syn, because the kinetic rate of refolding is slower than for WT. Since cardiolipin at this molar ratio is also an anchor for LC3 and an initiating signal for mitophagy[24,25], we next sought to determine whether LC3 binding to cardiolipin was influenced by the presence of α-syn, and vice versa.

**α-Syn and LC3 compete for binding to cardiolipin.** Our final set of binding experiments evaluated the impact of LC3 on the binding of WT, A53T or E46K α-syn to OMM vesicles containing

**Fig. 4** Cardiolipin is externalized to the mitochondrial surface in α-syn mutant hNs and transgenic mice in response to α-syn accumulation. **a** Western analysis of lysates from corrected and A53T cells at DIV 14 and DIV 60 labeled for total α-syn, PS129, TH (DA neuronal marker), nestin (NPC marker) or GAPDH show that levels of PS129-modified α-syn are elevated in A53T hNs at DIV 14 and remain elevated at DIV 60 relative to corrected cells. **b, c** Mitochondria were purified from A53T cells and genetically corrected cells at both DIV 14 and DIV 60. Lysates were then separated based on TX-100 solubility (soluble) or urea solubility (insoluble). Western analysis of A53T cells shows elevated soluble α-syn at the mitochondria relative to corrected cells at DIV 14 and DIV 60 (denoted by ** and * respectively). Quantification of soluble and insoluble α-syn levels in mitochondrial fractions normalized to Coomassie (**c**). Data represent mean ± s.e.m. *$P < 0.05$, **$P < 0.01$ by ANOVA followed by Tukey's post hoc test, $n = 4$. **d, e** Micrographs of corrected and A53T-mutant NPCs expressing the negative charge probe RPRE-RFP and CL-GFP show RPRE-RFP translocation from a primarily plasma membrane localization in isogenic-corrected hNs to a punctate intracellular localization in A53T hNs in tight cellular localization with CL-GFP (**d**). Quantification of percent total cell number with colocalized RPRE-RFP and CL-GFP (**e**). Data represent mean ± s.e.m. **$P < 0.0001$ by $t$-test, $n = 3$ coverslips over 2 independent differentiations, DIV: 14; scale bar: 10 μm. **f, g** Fluorescence resonance energy transfer (FRET) from CL-GFP to RPRE-RFP in hiPSC-derived A53T and corrected hNs or hESC-derived WT, A53T and E46K hNs was assessed (**f**) and mean FRET intensity was quantified (**g**). Data represent mean ± s.e.m. **$P < 0.01$ by ANOVA followed by Tukey's post hoc test, for hiPSCs, $n = 6$, for hESCs, $n = 8$ coverslips over two independent differentiations, DIV: 14. Scale bar: 10 μm. **h** Mitochondria were isolated from the SNpc of 6-month-old WT and A53T transgenic animals and labeled with Annexin V to measure the abundance of cardiolipin on the mitochondrial surface. Controls were rotenone-exposed WT samples. Animals were not randomized, nor was the analysis blinded. Data represent mean ± s.e.m. **$P < 0.01$ by ANOVA followed by Dunnett post hoc test, $n = 4$. Clipart was obtained at clker.com

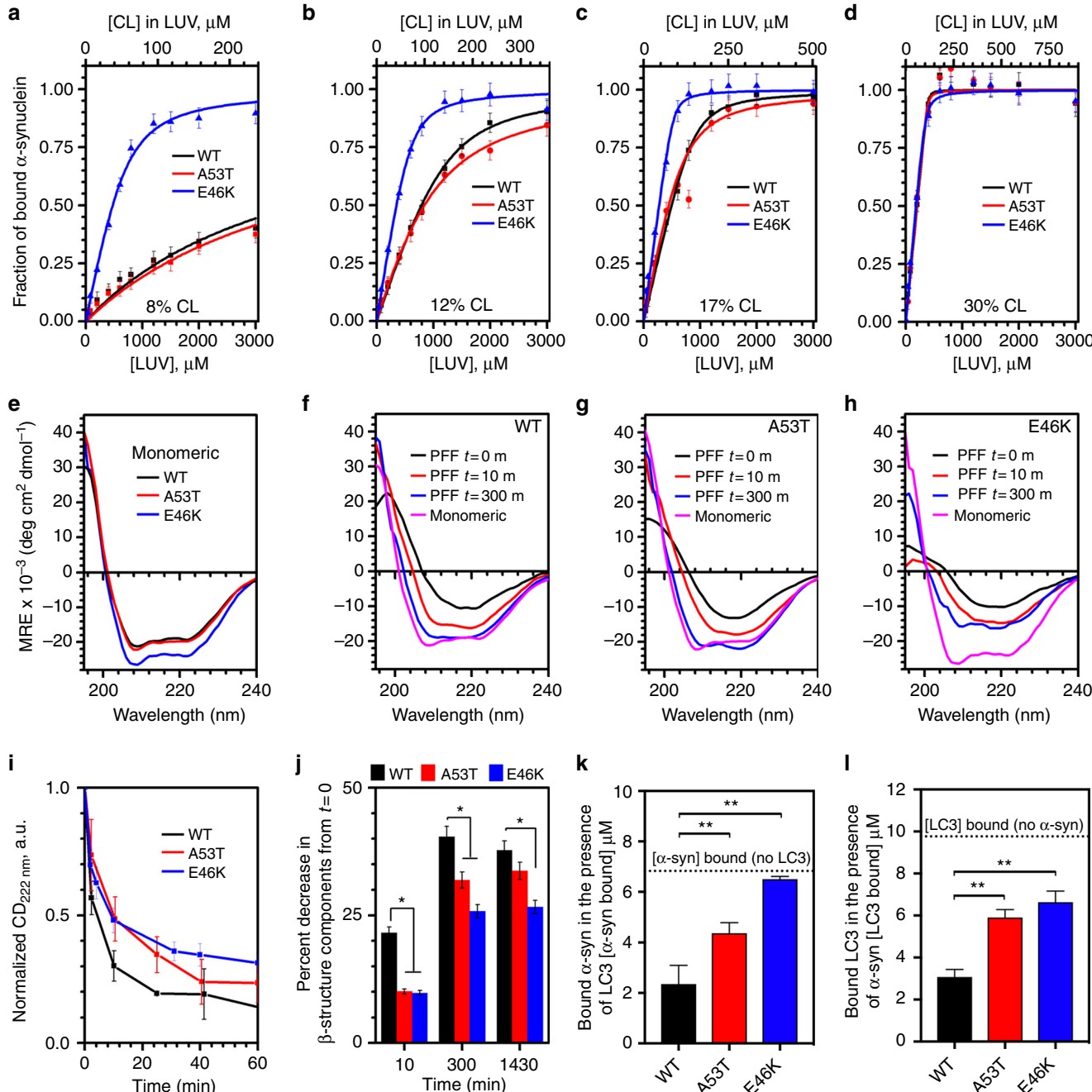

**Fig. 5** Cardiolipin promotes refolding of α-syn fibrils. **a–d** Binding of WT, A53T or E46K α-syn monomers to OMM-like LUVs, with increasing molar ratios of CL, was measured by CD spectroscopy and the 222 nm signal was used to calculate the fraction of bound α-syn. Data are plotted as change in fraction of bound protein vs. the total concentration of LUVs (bottom axis) or the total concentration of cardiolipin in the LUV (top axis). Binding parameters were calculated by fitting the data to Eq. 1 (Methods) and are reported in Supplementary Table 1. **e** Representative CD spectra of the monomeric WT, A53T or E46K α-syn in the presence of 1.2 mM LUVs containing 30% CL. **f–h** Refolding of WT (**f**), A53T (**g**) or E46K (**h**) α-syn preformed fibrils in the presence of 1.2 mM LUVs containing 30% CL was monitored at multiple time points by CD spectroscopy to follow the redistribution of the secondary structural components (from primarily β-structures to primarily α-helical structures). The magenta line shows the CD spectra of the monomeric WT, A53T or E46K α-syn in the presence of 1.2 mM LUVs as a reference standard. **i** Change in CD signal at 222 nm collected as a function of time. Kinetic data were fitted by exponential decay and the calculated rates of refolding were $3.79 \pm 0.46\ \mathrm{h}^{-1}$ vs. $2.44 \pm 0.42\ \mathrm{h}^{-1}$ vs. $1.30 \pm 0.49\ \mathrm{h}^{-1}$ for WT, A53T and E46K α-syn, respectively ($P < 0.05$). **j** Relative decrease in the β-structure components determined by deconvolution of CD spectra presented in **f–h**. Data represent mean ± s.e.m. *$P < 0.05$ by ANOVA, followed by Dunnett post hoc test, $n = 3$. **k**, **l** WT, A53T or E46K α-syn (10 μM) was incubated with 0.5 mM LUVs containing 30% cardiolipin in the presence or absence of equimolar LC3 protein and concentration of α-syn bound to LUVs (**k**) or LC3 bound to LUVs (**l**) was determined by flotation assay. Pleated line indicates the amount of LUV bound α-syn in the absence of LC3 (**k**) or the amount of LC3 bound in the absence of α-syn (**l**). Data represent mean ± s.e.m. **$P < 0.01$ by ANOVA, followed by Dunnett post hoc test, $n = 3$

30% cardiolipin. Equimolar ratios of recombinant human LC3 were incubated with either WT, A53T or E46K α-syn in the presence of cardiolipin containing OMM vesicles and flotation assays were performed to assess binding of each respective protein to OMM vesicles. We found that LC3 significantly inhibited binding of WT α-syn to OMM vesicles (Fig. 5k) relative to WT α-syn alone (Fig. 5k, pleated line), but that LC3-mediated inhibition was significantly reduced in cases of A53T and E46K α-syn, having no impact on E46K binding whatsoever. Moreover, we found that WT α-syn similarly inhibited LC3 binding to OMM vesicles relative to LC3 α-syn alone (Fig. 5l, pleated line), suggesting that the two proteins are not normally present on OMMs simultaneously. The inhibition of LC3 binding to OMM vesicles was also significantly reduced in the cases of A53T and E46K α-syn. These data suggest that WT α-syn can interact with cardiolipin at the mitochondria without triggering LC3-dependent clearance of mitochondria, as WT α-syn inhibits LC3 binding to OMM vesicles. Mutant α-syn variants, however, do not share this inhibitory effect on LC3 binding and in fact co-exist on cardiolipin-rich OMMs. This finding would explain, at least in part, why *SNCA*-mutant hNs manifest increased mitochondrial clearance relative to control hNs.

**SNCA-mutant hNs transmit mitochondrial pathology.** Several recent studies have shown that aberrantly folded α-syn can transmit to and seed synucleinopathy in healthy cells, propagating the pathology[31–33]. These reports support a prion-like hypothesis of PD pathogenesis. Indeed, pathological analysis of graft tissue from a human PD subject who received a transplant of embryonic mesencephalon revealed Lewy body-like pathology in the grafted tissue 14 years after transplant[34].

In order to assess whether misfolded α-syn, and consequently mitochondrial dysfunction, could be propagated from A53T to corrected cells, we developed a co-culture system in which isogenic-corrected cells expressing GFP were co-cultured with A53T cells at DIV 14 and differentiated together. Differentiation in co-culture had no impact on the proportion of TH[+ve] neurons generated (Supplementary Fig. 6a–b). We labeled co-cultured GFP[+ve]-corrected and GFP[−ve]-A53T hNs for α-syn PS129 and ubiquitin to assess early signs of synucleinopathy. We noted accumulation of both of these pathological markers in GFP[+ve] and GFP[−ve] hNs (Fig. 6a) and quantification of α-syn PS129[+ve] cells showed similar elevated levels in both GFP[+ve] and GFP[−ve] populations (Fig. 6b).

To assess the impact of co-culture on mitophagic clearance, we next combined isogenic-corrected cells expressing GFP and MitoDSRed with A53T-mutant cells expressing MitoDSRed alone and differentiated them in co-culture. Upon terminal differentiation of co-cultured cells, we antigenically labeled cells with the autophagy marker LC3 and assessed formation of mitochondria-localized LC3 puncta in both GFP[+ve] and GFP[−ve] neurites (Fig. 6c). As noted above, A53T hNs had significantly higher numbers of cells containing fragmented mitochondria (Fig. 2) and a greater number of LC3[+ve] puncta (Fig. 3). In contrast, analysis of the co-cultured neurons showed that GFP[+ve]-corrected hNs accumulated LC3[+ve] puncta to the same extent as GFP[−ve] mutant hNs, and that these puncta were localized to mitochondria (Fig. 6c). Furthermore, both the GFP[−ve] mutant hN and the GFP[+ve]-corrected hN cultures contained similar proportions of cells with fragmented mitochondria (Fig. 6d). We also confirmed that these events translated into loss of mitochondrial potential in co-cultured GFP[+ve] and GFP[−ve] hNs by flow cytometry of TMRE-labeled cells. In contrast to our previous experiments (Fig. 3e), which showed that corrected hNs maintained normal mitochondrial potential when differentiated alone, co-culture with A53T hNs resulted in depolarization of mitochondria in GFP[+ve]-corrected hNs (red trace) as indicated by a shift in TMRE fluorescence to levels resembling the GFP[−ve]-A53T population (black trace) (Fig. 6e and Supplementary Fig 6c). GFP[+ve]-corrected hNs co-cultured with untransformed (GFP[−ve]) corrected hNs showed no evidence of α-syn PS129 accumulation and no LC3 punta formation (Supplementary Fig. 6d, e).

We next assessed whether these effects could be attributed to cardiolipin translocation to the OMM and associated accumulation of anionic charge on the surface of mitochondria in corrected cells. We expressed both CL-GFP and RPRE-GFP in corrected cells (only) and differentiated them alone or in the presence of A53T cells. Whereas corrected cells cultured alone exhibited a plasma membrane localization of RPRE-RFP that is distinct from the CL-GFP expression pattern, co-culture with A53T cells caused redistribution of RPRE-RFP and resulted in colocalization of this negative charge probe with CL-GFP (Fig. 6f). This result indicates that corrected hNs undergo a change in mitochondrial surface charge corresponding to externalization of cardiolipin in the setting of co-culture with A53T hNs. We further characterized the proximity of RPRE to cardiolipin by FRET in either hiPSC-derived Corrected hNs, differentiated in co-culture with isogenic A53T hNs, or in WT hNs, differentiated in co-culture with isogenic A53T or E46K hNs. We found that co-culture with *SNCA*-mutant neurons resulted in a significant increase in the level of energy transfer between GFP-labeled cardiolipin and RFP-labeled RPRE in both corrected and WT hNs relative to when these lines are cultured alone (Fig. 6g, h), indicating that co-culture with *SNCA*-mutant neurons alters cardiolipin localization within mitochondrial membranes of neighboring, non-mutant neurons. These data further suggest that co-culture of healthy neurons with *SNCA*-mutant neurons can (1) trigger the onset of synucleinopathy, (2) initiate cardiolipin externalization to the OMM and (3) promote LC3-dependent mitochondrial clearance. Collectively, these results suggest that molecular markers of PD pathology can be transmitted from mutant to healthy human cells in our system.

**Antibodies against α-syn prevent transmission of pathology.** Next, we sought to simultaneously test whether secreted α-syn was indeed responsible for the transmission of mitochondrial pathology from A53T to corrected hNs as well as whether a specific α-syn mAb could prevent the spread of disease by blocking α-syn transmission and rescuing mitochondrial pathology. For these experiments, we co-cultured hiPSC-derived, GFP-labeled isogenic-corrected and unlabeled A53T α-syn cells from DIV 14 to DIV 60 in the presence of either $1 \, \mu g \, ml^{-1}$ α-syn mAb (anti-α-syn) or $1 \, \mu g \, ml^{-1}$ of a non-specific IgG. An identical experimental paradigm was established in hESC-derived hNs by co-culturing GFP-labeled WT cells with isogenic A53T or E46K *SNCA*-mutant cells. The efficiency of α-syn capture from media is reported in Supplementary Table 2. We then labeled the co-cultures at DIV 60 for α-syn PS129. Importantly, the presence of anti-α-syn mAb compared to IgG significantly reduced the percentage of α-syn PS129[+ve]/GFP[+ve]-corrected hNs relative to total GFP[+ve]-corrected hNs (Fig. 7a, b and Supplementary Fig. 7a, b). In contrast, the presence of anti-α-syn mAb compared to IgG had no effect on the proportion of α-syn PS129[+ve]/GFP[−ve]-*SNCA*-mutant hNs. This result is consistent with the notion that α-syn-mAb present in the media captures secreted α-syn to inhibit the spread of pathology from the *SNCA*-mutant cells to the isogenic control cells within the co-culture system.

Finally, we examined the impact of blocking α-syn transmission on the propagation of mitochondrial dysfunction in co-

cultured control and *SNCA*-mutant hNs from both hiPSC and hESC backgrounds. GFP/mitoDSRed co-expressing control cells were again co-cultured with *SNCA*-mutant cells expressing mitoDSRed in the presence of either $1\,\mu g\,ml^{-1}$ α-syn-mAb (anti-α-syn) or $1\,\mu g\,ml^{-1}$ of non-specific IgG or vehicle from DIV 14 to DIV 60. Whereas GFP$^{+ve}$ control hNs of hiPSC (Fig. 7c, d) and hESC (Fig. 7e, f) origin cultured alone had intact mitochondria, co-culture of GFP$^{+ve}$ control hNs with *SNCA*-mutant hNs in the presence of IgG resulted in mitochondrial fragmentation (Fig. 7c–f). This effect was rescued, in part, by treatment with anti-α-syn (Fig. 7c–f). These results support a critical role for secreted α-syn in the transmission of mitochondrial pathology in human PD neurons and suggest that α-syn mAbs may effectively block both proteostatic and mitochondrial stress-induced cellular phenotypes in PD.

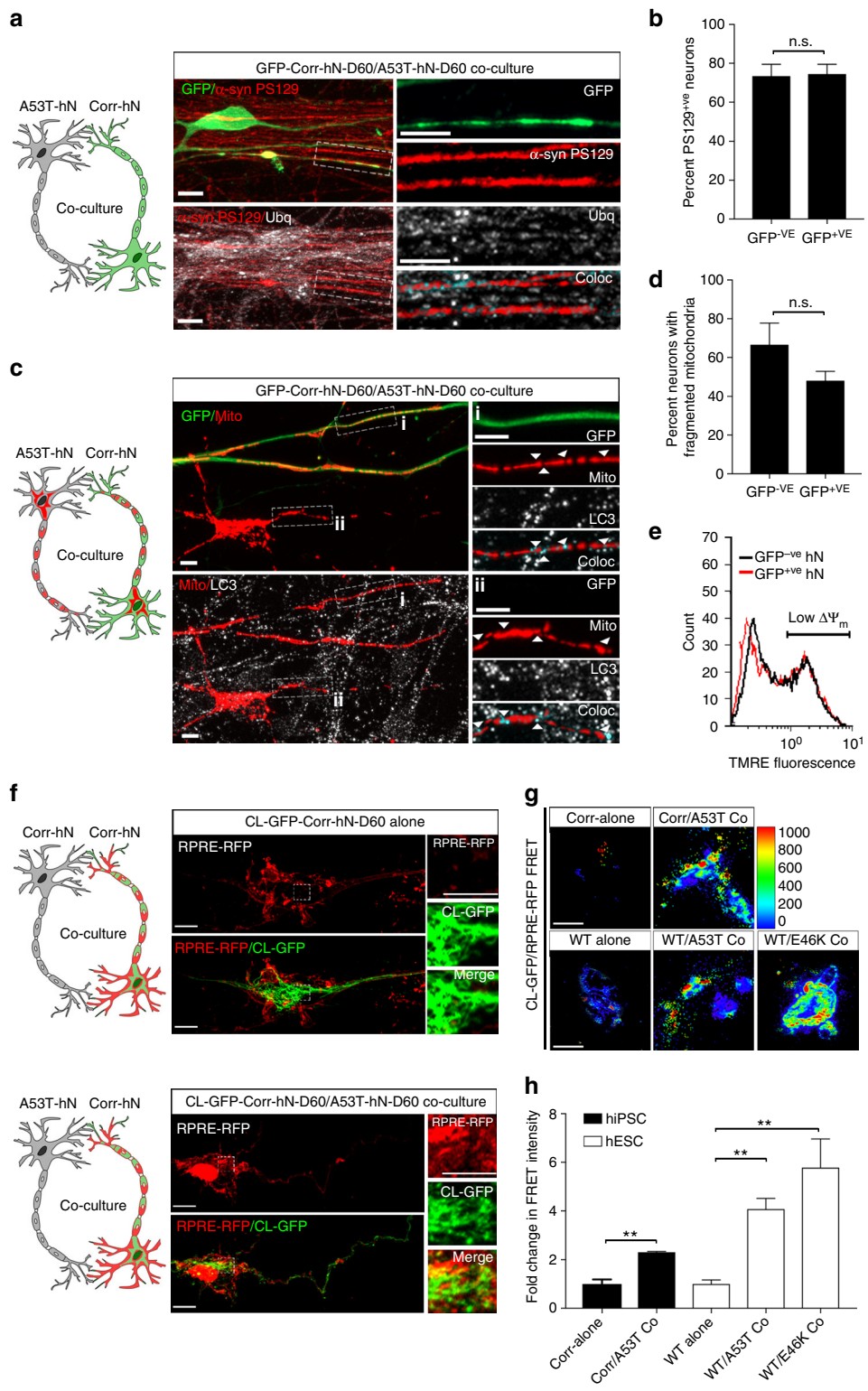

## Discussion

Although mitochondrial function is known to be aberrant in PD, the link between this dysfunction and the deposition of toxic protein aggregates has remained unclear. In this study, we describe a novel mechanism in which cardiolipin plays a vital role in α-syn folding. Furthermore, we show that OMM-localized cardiolipin can pull α-syn monomers out from preformed fibrils, facilitating their refolding, thus effectively buffering synucleinopathy. This buffering capacity is reduced by mutations in α-syn that cause familial PD. Finally, we show that mitochondrial pathology can spread from human α-syn mutant cells to non-mutant cells by direct transmission of mutant α-syn. Importantly, this spread can be blocked by an anti-α-syn mAb.

In animal models of both familial and sporadic neurodegenerative conditions, including PD, morphologically abnormal mitochondria appear in the brain as a result of altered fission/fusion dynamics[35–37], with α-syn reportedly driving mitochondrial fission[14] and inhibiting fusion[38]. Although α-syn has been reported to interact with several types of intracellular membranes[39], it appears to have a preferential affinity for mitochondrial or mitochondria-associated membranes[15,40]. The initial attraction of α-syn to these membranes seems to require anionic charge, as in vitro quenching of the cardiolipin phosphate group in artificial mitochondrial membranes directly inhibits the interaction between α-syn and the mitochondrial membrane[41]. The physical interaction of WT α-syn with cardiolipin has been reported to involve the N-terminal region of WT α-syn and the cardiolipin acyl side chains[42]. The acyl chains induce negative curvature strain in cardiolipin microdomains which, coupled to the divalent anionic charge from the diphosphatidyl glycerol headgroup, has been proposed to facilitate docking of α-syn oligomers[43,44]. The LC3 binding to cardiolipin has also been reported to be dependent on the presence of the acyl side chains, as LC3 binding to lyso-cardiolipin is dramatically reduced[25]. Our data suggest that WT α-syn and LC3 compete for binding to the same site on cardiolipin, identifying a possible mechanism by which α-syn may regulate LC3-induced mitophagy. A53T and E46K α-syn had a significantly reduced ability to competitively inhibit LC3 binding to cardiolipin, which may explain the increase in mitophagic flux observed in SNCA-mutant neurons.

It has been proposed that anionic phospholipids, such as cardiolipin, can promote folding of disordered α-syn monomers to an α-helical structure[14,45]. In support of this hypothesis, we are the first to show that cardiolipin has the capacity to buffer synucleinopathy by drawing α-syn monomers out of potentially pathological oligomers and fibrils, a capacity that is reduced in fibrils composed of mutant α-syn. This observation is in contrast to reports that anionic membranes promote α-syn aggregation[46–48]; however, these early studies did not focus on cardiolipin-containing mitochondrial membranes, but rather on synaptosomes and phosphatidic acid-containing LUVs, suggesting that the observed effects on refolding of α-syn fibrils may be unique to cardiolipin-containing mitochondrial and mitochondria-associated membranes. Our findings therefore establish a novel role for mitochondria in the proper folding and refolding of α-syn and further suggest that mitochondria may be an important defense mechanism against abnormal α-syn deposition and Lewy body formation.

Based upon our results, it is tempting to speculate that the early-onset PD phenotype associated with the A53T or E46K mutations may be related, at least in part, to the reduced capacity of the mitochondria to refold mutant α-syn. The increased abundance and duration of cardiolipin exposure on the OMM needed to refold mutant α-syn would alter membrane dynamics and may initiate the depolarization of mitochondrial membranes and associated mitochondrial stress that we observe in SNCA-mutant neurons. Recent reports that oxidatively modified α-syn interacts with Tom20 and impairs mitochondrial protein import[49] may lend insight into the mechanism by which α-syn association with the mitochondria facilitates cardiolipin exposure. It is possible that oligomeric α-syn can similarly impair mitochondrial protein import and that this signals translocation of cardiolipin from the IMM to the OMM to alleviate this inhibition.

Multiple lines of evidence have shown that oligomeric α-syn aggregates can seed pathology in previously healthy cells by promoting oligomeric aggregation[32,50,51]. Striatal injections of WT α-syn PFFs in healthy mice result in the accumulation of hyperphosphorylated α-syn deposits in other brain regions with a direct connection to the striatum within 30 days, suggesting that cell-to-cell transmission occurs in rodents and that it follows inter-neuronal connectivity[32]. Our data are, to our knowledge, the first to show that endogenous mutant α-syn can transmit cell to cell in a hPSC-derived neuronal system, triggering mitochondrial pathology in healthy cells. Moreover, these effects manifest within months, which supports the similarly short timeframe for transmission observed in mice[32]. In addition, our findings highlight and support the importance of recent advances in α-syn immunotherapy, in which mAbs against misfolded α-syn reduced the formation of Lewy bodies and Lewy neurites in mice injected with preformed fibrils[50]. Our results show that an α-syn antibody

**Fig. 6** A53T hNs transmit mitochondrial pathology to corrected hNs. **a**, **b** Micrographs of co-cultured GFP$^{+ve}$-corrected and GFP$^{-ve}$-A53T-mutant hNs antigenically labeled for ubiquitin (Ubq) and PS129; scale bar: 10 μm (**a**). Enlarged regions of GFP$^{+ve}$ neurites and GFP$^{-ve}$ neurites show that both display equal Ubq/PS129 colocalization. Boolean operation was employed to label regions of colocalization in blue, scale bar: 50 μm. Quantifications of PS129$^{+ve}$ neurons within the GFP$^{+ve}$ and GFP$^{-ve}$ populations (**b**). Data represent mean ± s.e.m. $P = 0.8919$ by t-test, 10 coverslips over 3 independent differentiations, DIV: 60. **c**, **d** Micrographs of GFP$^{+ve}$-corrected and GFP$^{-ve}$-A53T-mutant hNs expressing mitoDSRed and antigenically labeled for endogenous LC3. GFP$^{+ve}$ neurites (**i**) and GFP$^{-ve}$ neurites (**ii**) are enlarged and show LC3 punctate colocalizes with mitochondria. Arrows show areas where LC3 colocalizes with MitoDSRed. Boolean operation was employed to label regions of colocalization in blue, scale bar: 10 μm. **d** Quantification of percentage of total GFP$^{+ve}$ and GFP$^{-ve}$ hNs that have fragmented mitochondria. Data represent mean ± s.e.m. $P = 0.1479$ by t-test, $n = 8$ coverslips over 3 independent differentiations, DIV: 60. **e** Co-cultures were labeled with 200 nM TMRE, and mitochondrial potential ($\Delta \Psi_m$) was measured by flow cytometry using the dequench method in GFP$^{+ve}$ (red trace) and GFP$^{-ve}$ (black trace) populations. Representative plot illustrating that GFP$^{+ve}$ corrected cells within co-cultures have adopted a pattern indicative of mitochondrial depolarization. Representative trace from 3 independent experiments (10,000 events per experiment), DIV: 60. **f** Corrected cells expressing CL-GFP and RPRE-RFP were differentiated in co-cultured with either RFP$^{-ve}$/GFP$^{-ve}$ Corrected or RFP$^{-ve}$/GFP$^{-ve}$ A53T hNs. Translocation of cardiolipin from the IMM to the OMM was assessed by visualizing the redistribution of RPRE-RFP from the plasma membrane to the cardiolipin-GFP$^{+ve}$ mitochondria. Magnified insets show CL-GFP colocalization with RPRE-RFP in Corrected hNs when differentiated in co-culture, scale bar: 10 μm. **g**, **h** Fluorescence resonance energy transfer (FRET) from CL-GFP to RPRE-RFP in hiPSC-derived A53T and corrected hNs or hESC-derived WT, A53T and E46K hNs was assessed (**g**) and mean FRET intensity was quantified (**h**). Data represent mean ± s.e.m. **$P < 0.01$ by Student's t-test, $n = 5$ coverslips over two independent differentiations, DIV: 60. Scale bar: 10 μm. Clipart was obtained at clker.com

is also effective in blocking mitochondrial pathology that results from α-syn transmission in a human context using hPSC-derived A9-type DA neurons.

Multiple mechanisms have been proposed to explain the prion-like transmission of α-syn, from tunneling nanotubes that facilitate trafficking of α-syn-containing lysosomes[52] to direct excretion of α-syn due to lysosomal perturbations[53] or direct exo- and endocytosis of α-syn via exosomes[54]. Indeed, these mechanisms are likely not mutually exclusive. We detected a 14 kDa α-syn species in conditioned media from hPSC-derived neurons suggesting at least some α-syn is freely secreted. Moreover, we show that mAbs targeting α-syn can reduce, in part, the

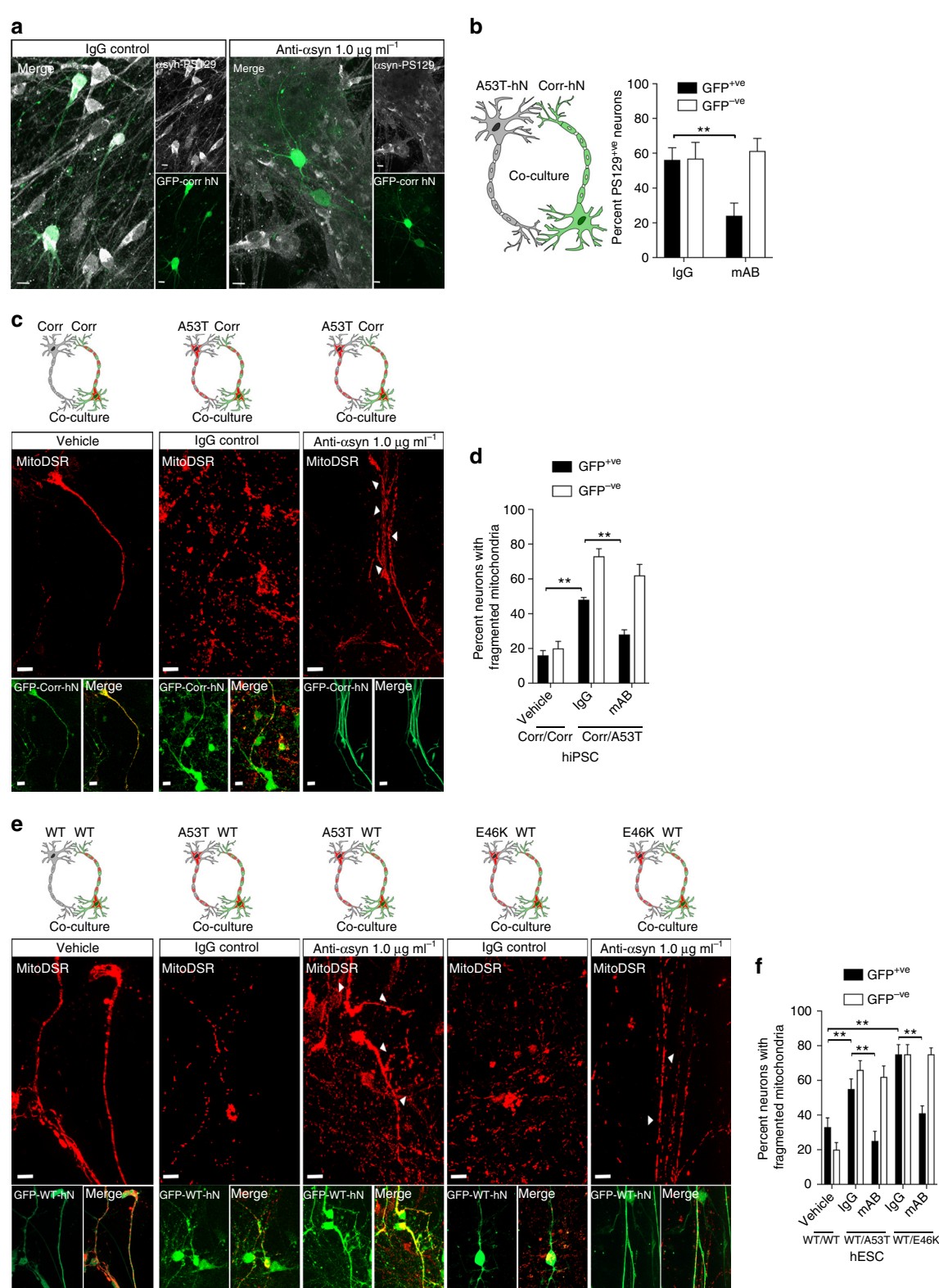

transfer of α-syn pathology between cells, suggesting that the C-terminal epitope of α-syn remains available. However, this would not rule out concurrent exosome-mediated transfer of α-syn, and may even include α-syn transferred via exosomes as the nature of the interaction between exosomal vesicles and α-syn is not well characterized. This membrane interaction may involve the α-syn N terminus (as described for mitochondrial membranes), which may still permit C-terminal capture via mAbs if exposed on the membrane surface. Moreover, we also detect α-syn in association with lysosomes, and therefore cannot rule out the simultaneous transfer of α-syn through multiple mechanisms. Nonetheless, the data presented here and the work of others suggest that α-syn immunotherapy is a promising avenue that warrants further exploration and that therapeutic approaches that block α-syn transmission may have a role as an adjunctive therapy.

## Methods

**hiPSC/hESC culture**. With the exception of the WT BGO1 hESCs, the cell lines used in this study were generated and kindly shared by Dr. Rudolf Jaenisch[16]. BGO1 hESCs were derived by BresaGen Inc. Genotypes of WT/Corrected and A53T cell lines were confirmed by restriction digest of genomic DNA, as previously described[16]. hPSCs (hiPSCs/hESCs designated as PSCs hereafter) were routinely cultured and maintained in our laboratory using a protocol described previously[12] with slight modifications. Briefly, pluripotent cells were plated on γ-irradiated human foreskin fibroblasts and cultured using ESC medium containing 20% knockout serum replacement (KSR) and 8 ng ml$^{-1}$ basic fibroblast growth factor (FGF), changed daily. The colonies were manually passaged weekly.

**A9 DA neuronal differentiation**. Differentiation of PSCs into A9-type DA neurons was performed as described previously[12,17]. Immediately preceding differentiation, the colonies were dissociated into a single cell suspension using accutase. To purify PSCs and remove fibroblast feeders, medium containing dissociated fibroblasts and PSCs was placed in gelatin-coated dishes. After adherence of the dissociated fibroblasts, supernatant containing purified PSCs was collected and re-plated at $4 \times 10^4$ cells per cm$^2$ on Matrigel (BD)-coated tissue culture dishes for differentiation. Floor-plate induction was carried out using medium containing KSR, LDN193189 (100 nM), SB431542 (10 μM), Sonic Hedgehog (SHH) C25II (100 ng ml$^{-1}$, Purmorphamine (2 μM), FGF8 (100 ng ml$^{-1}$ and CHIR99021 (3 μM). On day 5 of differentiation, KSR medium was incrementally shifted to N2 medium (25%, 50%, 75%) every 2 days. On day 11, the medium was changed to Neurobasal/B27/Glutamax supplemented with CHIR. On day 13, CHIR was replaced with brain-derived neurotrophic factor (BDNF; 20 ng ml$^{-1}$), ascorbic acid (0.2 mM), glial-derived neurotrophic factor (GDNF; 20 ng ml$^{-1}$), transforming growth factor-β3 (TGFβ3; 1 ng ml$^{-1}$), dibutyryl cAMP (dbcAMP; 0.5 mM) and DAPT (10 μM) for 9 days. On day 20, cells were dissociated using accutase and re-plated under high cell density $4 \times 10^5$ cells per cm$^2$ in terminal differentiation medium (NB/B27+BDNF, ascorbic acid, GDNF, dbcAMP, TGFβ3 and DAPT) or DAN Medium (DA Neuron) on dishes pre-coated with poly-ornithine (15 μg ml$^{-1}$)/laminin (1 μg ml$^{-1}$)/fibronectin (2 μg ml$^{-1}$). Nucleofection with mt-mKeima (addgene), RPRE-RFP or RPRE-GFP (Dr. Sergio Grinstein, Hospital for Sick Children, Toronto) and CL-GFP (Dr. Joaquin Madrenas, McGill University, Montreal), was performed between days 11 and 14 (hNPC) or days 20–25 of terminal differentiation using human stem cell nucleofector kit 1 (Lonza, VPH5012) in conjunction with the Amaxa Nucleofector 2b system (Lonza) according to the manufacturer's protocol. Stable MitoDSRed and or GFP expression was achieved by lentiviral infection at DIV 11 of differentiation. For mAB treatments, cells were treated with a mouse monoclonal IgG$_1$ targeted against the C terminus of α-syn or an isotype-specific control mAB (Santa Cruz). Knockdown of autophagy genes was achieved by lentiviral expression of shRNA IRES GFP constructs targeted against Becn1 gene

product or scrambled controls using the following sequences: *Becn1*: GGAGG-CAGTGGCGGCTCCTATTCCATCAA or CACCATGCAGGTG AGCTTCGTGTGCCAGC.

**Animals**. B6.Cg-Tg(THY1-SNCA*A53T)M53Sud/J mice (The Jackson Laboratory) and wild-type controls were housed in compliance with ethical regulations and experiments were approved by the Animal Care Committee of the Scintillon Institute for Biomedical and Bioenergy Research. Male and female animals between 6 and 8 months of age were used.

**Immunocytochemistry and fluorescence analysis**. Cells were fixed with 4% paraformaldehyde (PFA) for 20 min, washed 1× with phosphate-buffered saline (PBS), and blocked with 3% bovine serum albumin and 0.3% Triton X-100 in PBS for 30 min. Cells were incubated with primary antibody overnight, and the appropriate AlexaFluor (488, 555, 647)-conjugated secondary antibodies were used at 1:2000. Primary antibodies and dilutions were as follows. Tom20 and Tom40 (1:500) were obtained from Santa Cruz Biotechnology (Cat. no. sc-17764 and sc-11414); GIRK2 (1:1000) was obtained from Millipore (Cat. no. AB5200). Phosphoserine 129 (81A) (1:1000) LC3 (1:500), beta-III-Tubulin and Ubiquitin (1:500) were purchased from BioLegend (Cat. no. 825701, 827101, 801202, 840501), while total α-synuclein (1:500) was from BD Biosciences (Cat. no. 610787). Anti-TH (1:1000) was from Pelfreeze (Cat. no. P40101) while anti-vGlut (1:500) and anti-GAD65 (1:500) were from Synaptic Systems (Cat. no. 135302, 198111). Cells were counterstained with 4',6-diamidino-2-phenylindole (1:500) from Invitrogen. For FRET analysis cells were labeled with donor and acceptor fluorophores alone to establish bleed-through constants prior to co-labeling and analysis. Analysis was performed using the FRET analysis tool of Volocity 6.3 (PerkinElmer). For analysis of percent PS129-positive neurons containing fragmented, mitoDSRed-expressing hNs were fixed, stained for PS129 and imaged using Axio-observer LSM 800 with Airyscan or using light-emitting diode (LED)-based illumination and Optical Sectioning by structured illumination (Zeiss), and morphological analysis of PS129-positive and MitoDSR-positive particles was performed using the measurements tool of Volocity 6.3 (PerkinElmer). PS129-positive neurons were determined by establishing a threshold of staining intensity in Volocity's Advanced Measurements Tools. Mitochondria within cells that were positive for PS129 staining were scored as fragmented if >15% of the total mitochondrial volume per cell existed in particles with a volume <5 μm$^3$. Proximity ligation assay was performed using the DuoLink In situ Red detection platform with mouse and rabbit probes. Imaging was performed using either an Axio-observer LSM 800 with Airyscan for super-resolution imaging by structured illumination (Zeiss) or an Axio-observer Live-cell imaging microscope with LED-based illumination and Optical Sectioning by structured illumination (Zeiss). Objectives used were Plan-APO 40×/1.4 Oil DIC (UV) VIS-IR or Plan-APO 63×/1.4 Oil DIC M27.

**Mitochondrial structure and function**. For fragmentation analysis, MitoDSRed expressing hNs were imaged using Axio-observer LSM 800 with Airyscan or using LED-based illumination and Optical Sectioning by structured illumination (Zeiss), and volumetric analysis of mitochondrial particles was performed using the measurements tool of Volocity 6.3 (PerkinElmer). Mitochondria were scored as fragmented if >15% of the total mitochondrial volume per cell existed in particles with a volume <5 μm$^3$. Mitochondrial potential was assessed by analyzing TMRE fluorescence levels using the dequench method and based on the manufacturer's protocol on a Beckman-Coulter FC500 flow cytometer. For TEM, neurons were fixed with 4% PFA/2% Gluteraldehyde, counterstained with 1% osmium and 1% uranyl acetate in cacodylate buffer, embedded in LR White, sectioned and imaged on a FEI Tecnai G2 F20 Transmission Electron Microscope.

**Mitochondrial fractionation and protein analysis**. Isolation of intact mitochondria from cultured cells was completed using the Mito Isolation Kit (Thermo). Cytosolic soluble proteins were extracted using the mitochondrial isolation reagent C (CHAPS-based lysis buffer) that was a component of the kit. Isolated

**Fig. 7** Blocking α-syn transmission blocks mitochondrial pathology. **a**, **b** A53T cells were co-cultured with GFP-expressing isogenic-corrected cells at DIV 14 and differentiated together (DIV 60) in the presence of either monoclonal anti-α-syn or IgG control antibody. Micrographs depict antigenic labeling of PS129 in co-cultured GFP$^{+ve}$ (Corr) and GFP$^{-ve}$ (A53T) hNs and show that anti-α-syn decreased PS129 labeling in GFP$^{+ve}$ hNs. scale bar: 10 μm (**a**). Quantification of data (**b**). Data represent mean ± s.e.m. **$P < 0.0001$ by ANOVA followed by Tukey's post hoc test, $n = 9$, DIV: 60. **c**, **d** Co-culture of GFP$^{+ve}$, MitoDSRed$^{+ve}$ corrected hNs with either hiPSC-derived corrected hNs or A53T hNs (as schematically depicted) in the presence of either monoclonal anti-α-syn, IgG or Vehicle, scale bar: 10 μm (**c**). Quantification of the effect of monoclonal anti-α-syn on the percentage of total GFP$^{+ve}$ hNs that have fragmented mitochondria (**d**). Data represent mean ± s.e.m. **$P < 0.01$ by ANOVA followed by Tukey's post hoc test, $n = 8$ coverslips over 3 independent differentiations. **e**, **f** Co-culture of GFP$^{+ve}$, MitoDSRed$^{+ve}$ WT hNs with either hESC-derived WT, A53T or E46K hNs (as schematically depicted) in the presence of either monoclonal anti-α-syn, IgG or Vehicle, scale bar: 10 μm (**e**). Quantification of the effect of monoclonal anti-α-syn on percentage of total GFP$^{+ve}$ hNs that have fragmented mitochondria (**f**). Data represent mean ± s.e.m. **$P < 0.01$ by ANOVA followed by Tukey's post hoc test, $n = 10$ coverslips over 3 independent differentiations. Clipart was obtained at clker.com

mitochondria were lysed using Tris-buffered saline (TBS)+1% Triton X-100. Samples were sonicated using a 2 mm probe tip sonicator (5 s pulse, 40% amp, 30 s total time) and spun at $100,000 \times g$ for 45 min at 4 °C. Supernatant was removed and labeled "soluble fraction". Pellets were washed twice in corresponding lysis buffer, and resuspended in 8 M urea+8% SDS in TBS. Samples were allowed to sit at room temperature for 30 min and then stored at −20 °C until needed. Protease and phosphatase inhibitors (NaF, PMSF, NaV, aprotinin) were added to lysis buffers just before use. Protein concentration was determined using the Bio-Rad DC Protein assay. Samples were separated on 4-12% gradient Bis-Tris sodium dodecyl sulfate–polyacrylamide gel electrophoresis (SDS–PAGE) gel, and transferred onto 0.2 μm nitrocellulose membrane. Membranes were probed with the following primary antibodies: total α-synuclein anti-mouse (BD Trans, 1:1000, Cat. no. 610787), GAPDH anti-mouse (Thermo, 1:2,000, Cat. no. MA5-15738), Nestin anti-mouse (R&D, 1:500, Cat. no. MAB1259), TH anti-rabbit (Pel-Freez, 1:1000, Cat. no. P40101), TOM40 anti-rabbit (Santa Cruz, 1:500 Cat. no. 11414). Donkey anti-mouse (Thermo, 1:2000) and anti-rabbit (Thermo, 1:2000) horseradish peroxidase-conjugated secondary antibodies were used in a 1:2000 dilution. Clarity Western ECL Blotting Substrate (Bio-Rad) was used to visualize bands on blots. Anti-mouse (800) and anti-rabbit (700) Li-Cor infrared conjugated secondary antibodies were used at 1:1000. Bands were visualized on the Li-Cor Fc imaging platform. Uncropped blots can be found in Supplementary Fig. 8.

**Protein purification and α-synuclein PFF formation.** The plasmid pET21a containing human α-synuclein complementary DNA was purchased through Addgene (Plasmid 51486) and deposited by MJFF. Mutagenesis was subsequently performed to generate the A53T variant using the Q5 site-directed mutagenesis kit (NEB) with the following mutagenesis primers: for A53T forward: GCATGGTGTGaCAA-CAGTGGC, reverse: ACCACTCCCTCCTTGGTT; for E46K forward: CAAAAC-CAAAAAAGGCGTGGT, reverse: CTGCCGACATACAGAACAC. Mutant and non-mutant colonies from a freshly transformed BL21-CodonPlus (DE3)-RIPL competent cells (Life Technologies) were used to inoculate a 10 ml starter culture, which was grown overnight at 37 C in an incubator shaker. Cell culture from the overnight growth was then diluted 100-fold and induced with 250 μg ml⁻¹ IPTG at A600 = 0.6 for 3 h. All the cultures were performed in LB media with 50 μg ml⁻¹ ampicillin at 37 °C. Cultures were centrifuged at 6000 rpm for 10 min at 4 °C and pellets were frozen at −80 °C until they were ready to be used. Protein purification was completed via boiling according to ref. [55] with slight modifications made to adjust for the anionic properties of α-synuclein. Briefly, the cell pellet from 500 ml culture was thawed on ice for 20 min, resuspended in 50 ml water, placed in a beaker of boiling water for 20 min with shaking every 5 min and then placed on ice for another 5 min. The lysate was centrifuged at $50,000 \times g$ for 30 min at 4 °C. Supernatant was transferred to a new tube, Tris-HCl (pH 8.0) was added to a final concentration of 20 mM and supernatant was run through a 0.2 μm filter. Anion-exchange chromatography (FPLC) was performed on a 1 ml Pall AcroSep column connected to a DuoLogic system (Bio-Rad). Fractions were analyzed using SDS–PAGE, and those fractions containing α-synuclein were pooled and run through a Symmetry 300, C18, reversed-phase high-performance liquid chromatography (HPLC) column on a Waters system with Millennium 32 software. After reversed-phase HPLC, the pure protein solution was frozen at -80 °C and lyophilized until dry. Samples were stored at -20 °C until further use. The A53T and E46K sequences was validated via mass spectrometry. Preparation of PFFs was completed as per ref. [56] with minor changes. The starting 30 mg ml⁻¹ monomeric stock solution was made fresh from lyophilized protein powder and not a frozen stock solution. The stock solution was then spun at $100,000 \times g$ at 4 °C for 60 min, and the supernatant was removed to produce PFFs. The supernatant was diluted in sterile PBS to a final volume of 500 μl and a final concentration of 5 mg ml⁻¹. Samples were put on a Vibromixer placed inside a 37 °C warm room and shaken for 7 days at 1000 rpm. A sedimentation assay was completed to confirm PFF formation, and PFFs were aliquoted (10 μl) and stored at −80 °C until needed. The plasmid (pET15b) containing human LC3B was purchased through Addgene (plasmid 73949) and deposited by Dr. Dieter Willbold's lab. The plasmid was transformed and expression induced in the same manner as α-synuclein. LC3B was purified as per the Willbold lab protocol[57].

**LUV preparation.** All lipids (porcine brain phosphatidylcholine (PC), porcine brain phosphatidylethanolamine (PE), bovine liver phosphatidylinositol (PI), porcine brain phosphatidylserine (PS) and cardiolipin (CL)) were purchased from Avanti Polar Lipids (Alabaster, AL) and stored at −20 °C until use. Aliquots of chloroform solutions of PI, PC, PS, PE and CL were combined in the following six molar ratios: (1) 9.4:55.2:7.3:28.1:0; (2) 8.8:51.9:6.9:26.4:6; (3) 8.6:50.8:6.7:25.9:8; (4) 8.3:48.6:6.4:24.7:12; (5) 7.8:45.8:6.1:23.3:17; and (6) 6.6:38.6:5.1:19.7:30. Large unilamellar vesicles with a 100 nm diameter, referred to as LUVs, were formed as previously described[58].

**CD spectroscopy.** All CD data were collected using a Jasco J-815 spectropolarimeter (Japan Spectroscopic, Tokyo, Japan) using a quartz cuvette with a 1 mm path length (Hellma, Concord, ON, Canada), with thermostatting at 37 °C using a Jasco PTC-424S/15 Peltier temperature controller (Japan Spectroscopic, Tokyo, Japan). The samples contained 4 μM of protein (WT, A53T or E46K α-synuclein) in 10 mM potassium phosphate buffer (pH 7.4) and various concentrations of LUVs. Each spectrum was collected at a scan rate of 100 nm min⁻¹ and represents an average of 3 scans. Corresponding buffer scans were subtracted from sample scans before analysis and presentation of the data. Secondary structure was analyzed using the CONTIN-LL algorithm[59] at Dichroweb[60]. Experiments were typically repeated two or more times with independently prepared samples. For kinetic and binding experiments, each CD value at 222 nm was acquired for 45 s with 1 s intervals and all data points were averaged.

**Analysis of cardiolipin binding.** In the binding experiments, the observed CD signal at 222 nm consists of a signal from two populations of protein molecules, bound and free, and therefore the fraction of bound protein can be expressed as $(CD_{obs}−CD_f)/(CD_b−CD_f)$ and fitted to the following equation using OriginPro 8 (Origin Lab Corporation, Northampton, MA):

$$\frac{CD_{obs}−CD_f}{CD_b−CD_f} = \frac{K_D+n[\alpha−syn]_t+[LUV]_t−\sqrt{\left(K_D+n[\alpha−syn]_t+[LUV]_t\right)^2−4n[\alpha−syn]_t[LUV]_t}}{2n[\alpha−syn]_t}, \tag{1}$$

where $CD_{obs}$ is the signal acquired at 222 nm after buffer subtraction; $CD_f$ is signal acquired at 222 nm in the absence of LUVs (signal of free protein); $CD_b$ is the signal acquired at 222 nm in the presence of saturating concentration of LUVs (signal of bound protein); $[\alpha−syn]_t$ is the total concentration of α-syn in the mixture (4 μM); $[LUV]_t$ is the total concentration of LUVs added to the reaction mixture; $n$ is the stoichiometry of binding (number of lipid molecules per molecule of protein); and $K_D$ is the dissociation constant.

**Impact of LC3 on cardiolipin binding.** Flotation assay experiments were based on a protocol described by Anton et al.[61]. Each experiment contained 10 μM of protein (LC3B alone, WT α-syn alone, A53T α-syn alone, E46K α-syn alone or LC3B and WT, A53T or E46K α-syn) and 0.5 mM LUVs in 125 μl potassium phosphate buffer (KPB). Mixtures were incubated at 37 °C for 1 h, after which they were mixed with 175 μl of 2.4 M sucrose. Using a syringe, 400 μl of 0.8 M sucrose and 300 μl of 0.5 M sucrose were carefully layered in each tube. Samples were centrifuged at 90,000 rpm for 3 h at 4 °C using a MLA-130 rotor (Beckmann). Following centrifugations, fractions were collected using a syringe, starting with the bottom 250 μl, containing free protein. The remaining 750 μl was collected as apparently bound protein. Free protein was quantified by western blot for LC3B and slot blot for α-synuclein. For LC3B, samples were separated on a Tricine SDS–PAGE gel and transferred onto a 0.2 μm nitrocellulose membrane and probed with rabbit anti-LC3B (Abcam; 1:1000). For slot blots, fractions were applied to a 0.2 μm nitrocellulose membrane via slot blot apparatus (Bio-Rad) and probed with mouse anti-α-synuclein (BD; 1:1000). Donkey anti-rabbit-800 and donkey anti-mouse-700 Licor secondary antibodies were used in a 1:10,000 dilution. Membranes were imaged on a Licor-Fc and quantified using Image studio suite version 5.2 (Licor). Quantitative densitometry was normalized to free protein standards.

**Statistical analysis.** Data represent at least three independent experiments, presented as mean ± s.e.m. Statistical significance was ascertained by Student's $t$-test or two-way analysis of variance (ANOVA) with appropriate post hoc testing; $p < 0.05$ was considered significant. All data were analyzed using Prism7 (Graphpad Software Inc.). For data not fitting a normal distribution, non-parametric tests were used.

**Data availability.** All data generated or analyzed during this study are included in this published article (and its Supplementary Information files) or is available from the authors upon request.

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

## Acknowledgements

This work was supported in part by the Parkinson Society of Canada (2014-685 to S.D.R.), the Natural Sciences and Engineering Research Council of Canada (RG060805 and CRDPJ490841-15 to S.D.R., RG121541 to G.H.), the CRC Program (to G.H.), and NIH grants R01 NS086890 and P30 NS076411 (to S.A.L.). We thank Robert Harris and Keith Sherriff at the University of Guelph for technical assistance with TEM.

## Author contributions

S.D.R. was responsible for project oversight and design; T.R. conducted cell culture, microscopy and mitochondrial function experiments with assistance from R.J.-W., D.D.M. and M.G.S.; T.R. and C.L.C. performed subcellular protein fractionation; V.V.B., C.L.C. and K.M.H. performed recombinant protein purification and reconstitution, while V.V.B., K.M. H., and G.H. performed CD spectroscopy and analysis; T.R., V.V.B., S.A.L., R.A., G.H. and S.D.R. prepared and edited the manuscript.

## Additional information

**Competing interests:** The authors declare no competing financial interests.

