## [Peer Review File · Nature Communications]

Reviewers' comments:

Reviewer #1 (Remarks to the Author):

This is a wide ranging manuscript that follows up on a prior study describing human iPSC-derived neurons harboring the PD aSyn A53T mutation and corrected controls.

The manuscript initially shows evidence for accumulation of aSyn with ubiquitin near mitochondria in these cultures, that is increased in mutant cultures at d60.

Comment >> there should be some analysis of control/other intracellular membranes and compartments to confirm specificity of this finding.

Subsequent data shows evidence of increased mitochondrial fragmentation depolarization and associated increased cardiolipin on the surface.

In a second line of study, there is evidence that cardiolipin interacts with aSyn and promotes folding of oligomers, which might be reduced in the mutants. >>This is somewhat consistent with prior studies on cardiolipin and aSyn.

Finally a third line of study presents evidence for transcellular spread of mitochondrial pathology in co-culture that is partially blocked by a-synuclein antibodies.

Questions and comments:

>> does the antibody have an effect on mutant cells directly, without co-culture?

>> a vehicle control should also be included, in addition to isotype control.

>> what is the isotype of the control antibody and of the aSyn antibody?

-- an overall concern is that all the studies completely rely on 2 clones. This should be addressed either by analysis of more clones, or by "rescue" studies where aSyn is induced or blocked.

-- it can be argued that each of the three areas of study here are not completely novel aspects of aSyn analysis. The study as a whole is interesting nonetheless.

Reviewer #2 (Remarks to the Author):

Cardiolipin exposure on the outer mitochondrial membrane modulates α -synuclein proteostasis in hiPSC-derived Parkinson's Disease neurons by Ryan et al

Ryan et al use human iPSC-derived dopamine neurons to suggest that externalization of cardiolipin to the cytosolic leaflet of the outer mitochondrial membrane promotes mitochondrial fission and clearance and this process is exaggerated in neurons containing the A53T mutation of SNCA compared to gene corrected controls. Furthermore, they suggest that a-synuclein can propagate from mutant neurons to wild type cells. While the data is interesting, there are a few areas that should be improved. Specifically,

1. Throughout the paper: While the authors list n reasonably well (although there are some minor statistical concerns below), I am concerned that the true 'n' is really 1 control and 1 A53T line. Some independent confirm in additional clones or, even better, in an independent system such as transgenic mice, would be very helpful.

2. An additional general comment is that the authors effectively use a nested design in several experiments. For example, in figure 1b the n=6 coverslips are from 3 differentiations. Therefore, coverslips are technically not independent samples and it could be reasonably argued that the true n=3. More of a concern is with experiments such as 2b where n=30-60 cells, which are not independent experiments. Are the data still significant if only independent experiments are considered as true n? An alternative way to deal with this type of data is to use ANOVA with

coverslip as a variable within the design. Given the very low n, the authors should probably use dot plots or boxplots rather than bars with error bars.

3. The colocalization of pS129 and Ub in figure 1 seems very high –most synuclein in cells is not ubiquitylated. Can the authors confirm biochemically that synuclein is modified in the cells and show the difference in figure 1 more quantitatively?

4. In figure 2c,d it looks like the two images are really of different planes of mitochondria, with the corrected neuron example being along the longer plane and the mutant neuron being sectioned across the shorter plane of the mitochondria. Additional examples of images would be helpful.

5. That TMRE and LC3 (an autophagy marker, not mitophagy as stated in the text) correlate is perhaps not surprising. What is a bit unusual is that FCCP does not promote complete mitophagy in the control line in 2j; was depolarization complete in both clones in this experiment?

6. I am unclear as to the mechanism by which A53T synuclein results in mitochondrial damage. In figure 3, the authors state that in cells cardiolipin is externalized as a response to A53T synuclein but in figure 4 suggest that cardiolipin promotes 'refolding' of fibrils with a higher affinity for WT over A53T using recombinant approaches. However, the higher affinity for WT might also be predicted to raise cardiolipin on the OMM if synuclein stays there to any extent. The authors should develop more testable predictions from their model.

7. The upper band in the western blot in 3a is probably not synuclein – see *Neurosci Lett.* 2003 Oct 2;349(2):133-5. Controls for specificity (eg siRNA) for all antibodies is needed.

8. I was concerned about the assay in 3i. As presented there isn't the resolution to distinguish between IMM (where cardiolipin is normally found) and the OMM. It is notable that these are not FRET but counts of cells with 'colocalization'. I also wasn't clear if these are live or fixed cells, which might disturb CL localization. Overall, these results need confirmation with independent techniques.

9. Figure 5d is fairly important, but no indication of statistical significance is shown on the bar graph. Is this statistically significant, considering the points raised above about nested designs?

10. In figures 5 and 6, it would be important to show GFP-positive cells are not affected when placed in co-culture with A53T corrected GFP-negative neurons. There is a single cell seen in figure 6c but this control should be in all panels in these two figures.

11. In 5e, why are so many cells low MMP? The pattern is very different from that in 2k.

12. For the antibody experiments it would be important to show that the antibody depletes extracellular synuclein.

13. Minor – some aspects of language need correcting. I don't think the assay in figure 4 is really refolding so much as adoption of helical conformation from unfolded monomer in solution.

14. Minor – the authors also oversell their results in the abstract in p2. Whether linking mitochondria and proteostasis has been done in this paper is arguable but there is certainly no evidence that linking these two events would lead to better therapies.

Reviewer #3 (Remarks to the Author):

The premise of this work is to determine the mechanisms underlying alpha-synuclein's ability to interact with mitochondria and drive neuropathologies associated with Parkinson's disease (PD). The study utilizes a human pluripotent-derived A53T-SNCA DA neuronal model versus a genetically corrected cells previously generated by the authors' to control for potential confounds associated with genetic background. The authors' report that prolonged interaction between alpha-synuclein oligomers and cardiolipin (CL) located on the mitochondrial surface in this model results in lysosomal mitophagy. Furthermore, co-culture of these mutant cells with genetically corrected isogenic neurons results in transmission of mitochondrial pathology to these cells that is blocked by alpha-synuclein antibody suggesting non-cell autonomous transmission of mitochondrial phenotypes through alpha-synuclein to normal neurons. The techniques used to morphologically analyze these cells are all state-of-the art.

The authors state that the link between mitochondrial dysfunction and synucleinopathy remains

unclear. However there appear to be several articles which not only link the two, but also explicitly demonstrate a specific interaction between oligomeric alpha-synuclein complexes and the mitochondrial phospholipid cardiolipin (CL). Indeed in one publication (Zigoneunu et al., 2012), the authors report that the interaction specifically involves the KAKEGVVAAAE repeat region of the N-terminus of the alpha-synuclein protein and acyl cardiolipin side chains. Reported outcomes of this interaction include both increases in mitochondrial fragmentation and APL activation, although the precise mechanisms involved were not determined in these previous studies (Nakamura et al., 2011; Grey et al., 2011; Plotegher et al., 2014, Ghio et al., 2016). Additional published data demonstrates that alpha-synuclein oligomers are sequestered by lysosomes and can be transferred from neuron-to-neuron by tunneling nanotubes (TNTs) within these vesicles (Abounit et al., 2016). Other researchers have suggested that cell-to-cell transmission may involve either direct cellular exocytosis-endocytosis of alpha-synuclein protein or delivery via exosomes (e.g. Quek and Hill, 2017; Spencer et al., 2017). All of these studies should be discussed in the context of the authors' own findings.

Chu and her colleagues previously demonstrated that under conditions of mitochondrial stress, IMM CL translocates to the OMM ('cardiolipin exposure') and that this in turn lowers the OMM surface charge. CL externalization has also been reported to allow mitochondrial interaction with alpha-synuclein (Nakamura et al., 2011). The authors' demonstrate that in their A53T cell model, CL externalization in response to A53T at early developmental stages coincides with reduced OMM charge. They suggest based on their data that CL externalization is an early stress-induced response to presence of alpha-synuclein which allows its re-folding into a functional alpha-helical state (already suggested by Nakamura's previous work) and prevents subsequent aggregate formation, but that this is inhibited in the presence of A53T mutation which has lower binding affinity for CL. This appears to precede reduced MMP, fragmentation and mitophagy. As Chu has previously shown, CL also acts as an anchor for LC3 and this may explain why A53T results in increased mitophagy and elevated alpha-synuclein aggregation. This is a somewhat novel idea which would be bolstered by inclusion of evidence in OMM LUVs that LC3 binding increases in the presence of increased CL concentrations and that this is inhibited in the presence of WT alpha-synuclein to a greater degree than A53T. One aspect that is still missing in the current study is how alpha-synuclein mechanistically elicits mitochondrial stress to begin with resulting in subsequent CL externalization, since this latter process appears to be required for alpha-synuclein binding to the mitochondria.

The authors further demonstrate that co-culturing A53T cells with genetically corrected (Corr) neurons results in increases in detrimental mitochondrial phenotypes, demonstrating transfer of effects of mutant alpha-synuclein from diseased to non-diseased cells. This is blocked by alpha-synuclein Ab suggesting that transmission is via the protein itself (although this is not verified, see below). This is an important concept in terms of whether alpha-synuclein immunotherapy may also prevent mitochondrial defects associated with PD or whether cell-to-cell transmission occurs via other means including nanotubes or exosomes. The authors' findings should be discussed in the context of previous work showing direct alpha-synuclein protein cell:cell transmission versus other recent studies suggesting occurrence via other means (exosomes, nanotubes).

Specific comments:

- (1) Note that it is extremely difficult to see and therefore accurately interpret the data panels throughout the manuscript due to their small size. This should be replaced with larger images and include higher magnifications where not presented (some of those that are included are out-of-focus e.g. 1c, 2e, 3f, etc and should be replaced).
- (2) For Fig. 2a,b, sizes of what were designated as fragmented versus unfragmented mitochondria should be included—how were % determined? Also for how % axons containing fragmented mitochondria in Fig. 2e,f.
- (3) It is perplexing that no p-129-syn staining and very few fragmented mitochondria are present in the representative panels of control axons in Fig. 2e as a quantitative value of 25% is given for a-syn-129 localized with fragmented mitochondria in cells in 2f. In contrast, Fig. 1 suggests

presence of a-129 co-localizing with ubiquitin in the quantitation data graphs of the corrected neurons. Fig. 2g is difficult to see and requires some type of quantitation.

(4) It does not appear that soluble oligomers were present in mitochondrial fractions from either A53T or Corr cells (Fig. 3b). This appears in contrast to what has been previously published. It would be helpful if the entire gel was included here.

(5) What happens when labeled A53T cells are co-cultures with A53T cells? Are elevated levels of alpha-synuclein inclusions and mitochondrial phenotypes observed versus in Corr cells co-cultured with A53T?

(6) Levels of alpha-synuclein in the conditioned media from A53T versus Corr cells should be quantified +/- Ab to verify that observed non-cell homologous effects are truly due to its secretion from the former.

Reviewer #1 (Remarks to the Author):

This is a wide-ranging manuscript that follows up on a prior study describing human iPSC-derived neurons harboring the PD aSyn A53T mutation and corrected controls.

The manuscript initially shows evidence for accumulation of aSyn with ubiquitin near mitochondria in these cultures, that is increased in mutant cultures at d60.

Comment >> there should be some analysis of control/other intracellular membranes and compartments to confirm specificity of this finding.

This analysis has been added to the manuscript and can be found in Supplementary Fig 2a,b.

Subsequent data shows evidence of increased mitochondrial fragmentation depolarization and associated increased cardiolipin on the surface.

In a second line of study, there is evidence that cardiolipin interacts with aSyn and promotes folding of oligomers, which might be reduced in the mutants. >>This is somewhat consistent with prior studies on cardiolipin and aSyn.

Finally a third line of study presents evidence for transcellular spread of mitochondrial pathology in co-culture that is partially blocked by a-synuclein antibodies.

Questions and comments:

>> does the antibody have an effect on mutant cells directly, without co-culture?

>> a vehicle control should also be included, in addition to isotype control.

We see no significant impact of the mAB on mutant cells directly. The impact of the antibody on mutant cells as well as vehicle controls are now included in figure 7 and supplementary figure 7.

>> what is the isotype of the control antibody and of the aSyn antibody?

The α -syn antibody is a mouse monoclonal IgG1 that targets the C-terminal domain of α -syn. This information has been added to the supplementary materials and methods.

-- an overall concern is that all the studies completely rely on 2 clones. This should be addressed either by analysis of more clones, or by "rescue" studies where aSyn is induced or blocked.

We now include two isogenic hiPSC lines in addition to three isogenic hESC lines (five clones) and contrast the effects of two independent *SNCA* mutations (*SNCA*-A53T and *SNCA*-E46K) against isogenic WT or mutation corrected controls. Data from these lines has been included throughout the manuscript (Figures 1, 2, 3, 4, 6 & 7 and Supplementary figures 4 & 7).

-- it can be argued that each of the three areas of study here are not completely novel aspects of aSyn analysis. The study as a whole is interesting nonetheless.

We appreciate the Referee's interest in these findings.

Reviewer #2 (Remarks to the Author):

Cardiolipin exposure on the outer mitochondrial membrane modulates α -synuclein proteostasis in hiPSC-derived Parkinson's Disease neurons by Ryan et al

Ryan et al use human ipSC-derived dopamine neurons to suggest that externalization of cardiolipin to the cytosolic leaflet of the outer mitochondrial membrane promotes mitochondrial fission and clearance and this process is exaggerated in neurons containing the A53T mutation of SNCA compared to gene corrected controls. Furthermore, they suggest that a-synuclein can propagate from mutant neurons to wild type cells. While the data is interesting, there are a few areas that should be improved. Specifically, 1. Throughout the paper: While the authors list n reasonably well (although there are some minor statistical concerns below), I am concerned that the true 'n' is really 1 control and 1 A53T line. Some independent confirm in additional clones or, even better, in an independent system such as transgenic mice, would be very helpful.

We now include two isogenic hiPSC lines in addition to three isogenic hESC lines (five clones total) and contrast the effects of two independent *SNCA* mutations (*SNCA*-A53T and *SNCA*-E46K) against isogenic WT or mutation corrected controls. In addition, we include a transgenic mouse model of PD that confirms cardiolipin externalization to the mitochondrial surface in response to mutant α -syn accumulation *in vivo*. Data from these systems has been included throughout the manuscript (Figures 1, 2, 3, 4, 6 & 7 and Supplementary figures 4, 6 & 7).

2. An additional general comment is that the authors effectively use a nested design in several experiments. For example, in figure 1b the n=6 coverslips are from 3 differentiations. Therefore, coverslips are technically not independent samples and it could be reasonably argued that the true n=3. More of a concern is with experiments such as 2b where n=30-60 cells, which are not independent experiments. Are the data still significant if only independent experiments are considered as true n? An alternative way to deal with this type of data is to use ANOVA with coverslip as a variable within the design. Given the very low n, the authors should probably use dot plots or boxplots rather than bars with error bars.

As suggested, we have re-analyzed our data using coverslips (or cultures where coverslips were not the culture format) as a variable and repeated our statistical analysis and graphed these data as appropriate. We also included the comparison of multiple mutations from different isogenic lines.

3. The colocalization of pS129 and Ub in figure 1 seems very high –most synuclein in

cells is not ubiquitylated. Can the authors confirm biochemically that synuclein is modified in the cells and show the difference in figure 1 more quantitatively?

To confirm ubiquitylation of synuclein, we performed heat stable fractionation of synuclein from crude lysates and assessed ubiquitylation of synuclein by western blot analysis. These data can be found in Supplementary figure 6f.

4. In figure 2c,d it looks like the two images are really of different planes of mitochondria, with the corrected neuron example being along the longer plane and the mutant neuron being sectioned across the shorter plane of the mitochondria. Additional examples of images would be helpful.

We have added to figure 2 additional, zoomed out images of entire cells where multiple mitochondria can be seen simultaneously. We also include zoomed in examples of representative mitochondria. We also added three additional isogenic lines (WT, A53T and E46K) to this analysis to better convey the consistency of this finding.

5. That TMRE and LC3 (an autophagy marker, not mitophagy as stated in the text) correlate is perhaps not surprising. What is a bit unusual is that FCCP does not promote complete mitophagy in the control line in 2j; was depolarization complete in both clones in this experiment?

As the Referee rightfully points out, complete mitophagy was not observed following 180 minutes of FCCP exposure, which was the longest time point depicted in the original figure 2. We did observe complete mitophagy in both clones following 360 mins of FCCP exposure. We have therefore extended the study duration to 360 mins and added simultaneous tracking of MitoDSRed positive particles to show complete mitophagy. These data can be found in the new figure 3c and d. In addition, the description in the text of LC3 as a marker of mitophagy has been corrected to refer to LC3 as a marker of autophagy.

6. I am unclear as to the mechanism by which A53T synuclein results in mitochondrial damage. In figure 3, the authors state that in cells cardiolipin is externalized as a response to A53T synuclein but in figure 4 suggest that cardiolipin promotes 'refolding' of fibrils with a higher affinity for WT over A53T using recombinant approaches. However, the higher affinity for WT might also be predicted to raise cardiolipin on the OMM if synuclein stays there to any extent. The authors should develop more testable predictions from their model.

As suggested by the Referee, we have expanded our mechanistic investigation with respect to how mutant syn initiates mitochondrial dysfunction. We now show that WT α -syn and LC3 compete for binding to cardiolipin, identifying a mechanism by which α -syn may regulate LC3-induced mitophagy. In addition, we find that A53T and E46K α -syn have a significantly reduced ability to competitively inhibit LCB binding to cardiolipin, which we believe explains the increase in mitophagic flux observed in *SNCA*-mutant neurons. These data can be found in the new figure 5k and l. We further propose that the

increased abundance and duration of cardiolipin exposure on the OMM needed to refold mutant α -syn alters mitochondrial membrane dynamics and initiates the depolarization of mitochondrial membranes and the associated mitochondrial stress that we observe in *SNCA*-mutant neurons. This has been added to the discussion of the revised manuscript on page 24.

7. The upper band in the western blot in 3a is probably not synuclein – see *Neurosci Lett.* 2003 Oct 2;349(2):133-5. Controls for specificity (eg siRNA) for all antibodies is needed.

The primary arguments put forth in this study are based on changes in both the total levels of the α -syn protein, and the levels PS129-modified α -syn protein that accumulates in *SNCA*-mutant neurons. Under denaturing conditions, as shown in the former figure 3a (new figure 4a), most α -syn would be represented on a western blot in the 14kDa form. In light of the Referee's concern that higher bands may not be synuclein we have therefore removed all reference to the higher bands from the manuscript (both figures and text).

8. I was concerned about the assay in 3i. As presented there isn't the resolution to distinguish between IMM (where cardiolipin is normally found) and the OMM. It is notable that these are not FRET but counts of cells with 'colocalization'. I also wasn't clear if these are live or fixed cells, which might disturb CL localization. Overall, these results need confirmation with independent techniques.

We have performed a FRET analysis of cardiolipin interaction with the cytosolic charge probe (RPRE) as suggested by the Referee. We have also included three additional isogenic lines (WT A53T and E46K) as well as *SNCA*-A53T transgenic and WT animals in this analysis. Furthermore, we have added an independent analysis of cardiolipin externalization via Annexin V labeling of live, isolated mitochondria using the methods validated by Chu et. al 2013, *Nat Cell Biology*. These data are in the updated figure 4f, g and h.

9. Figure 5d is fairly important, but no indication of statistical significance is shown on the bar graph. Is this statistically significant, considering the points raised above about nested designs?

This statistical analysis has now been performed on a per coverslip basis. The means of these data sets do not differ significantly, meaning that healthy neurons do take on mitochondrial pathology to a similar extent to *SNCA*-mutant neurons when co-cultured. This statistical analysis is now shown in the associated bar graph in the new figure 6d.

10. In figures 5 and 6, it would be important to show GFP-positive cells are not affected when placed in co-culture with A53T corrected GFP-negative neurons. There is a single cell seen in figure 6c but this control should be in all panels in these two figures.

Analysis of WT or Corrected cells either alone or cultured with unmodified WT or Corrected cell is now shown for all comparisons in the old figures 5 and 6. These data can be found in the new figure 6, 7 and supplementary figure 6.

11. In 5e, why are so many cells low MMP? The pattern is very different from that in 2k.

Quantification of the percent cells with low mitochondrial potential from multiple flow cytometry experiments is now presented in Figure 3f and Supplementary Figure 6c in addition to the graphical representation of the forward and side scatter from flow cytometry experiments. As a result, the previous flow cytometry trace in figure 5e has been replaced with one that is more representative of the average of multiple independent flow cytometry experiments. These new data show that the number of A53T cells with low MMP (old fig 2k) is consistent with the number both A53T and corrected cell with low MMP when these cell lines are co-cultured (old fig 5e).

12. For the antibody experiments it would be important to show that the antibody depletes extracellular synuclein.

Data showing the amount of synuclein captured in conditioned media are now presented in supplementary table 2.

13. Minor – some aspects of language need correcting. I don't think the assay in figure 4 is really refolding so much as adoption of helical conformation from unfolded monomer in solution.

We thank the reviewer for pointing out that this figure is unclear. We did show adoption of the helical conformation from unfolded monomers in solution in the old figure 4 (panels e-g, magenta line). In addition to monomers, however, the other lines in each of these panels (black, red and blue) are showing time dependent adoption of helical conformation from pre-formed synuclein fibrils. We refer to this as refolding because this represents the transition from aggregates containing beta sheet conformations to monomers containing helical conformations, rather than from random coiled monomers to helical monomers. To clarify this, we have adjusted the text and separated the monomer data into a single panel (new figure 5e). Figure 5f-j, now refers to the effect of cardiolipin containing OMMs on re-folding of synuclein pre-formed fibrils, using monomers as a reference standard for adoption of the helical conformation.

14. Minor – the authors also oversell their results in the abstract in p2. Whether linking mitochondria and proteostasis has been done in this paper is arguable but there is certainly no evidence that linking these two events would lead to better therapies.

The abstract has been adjusted to better represent the results of our manuscript.

Reviewer #3 (Remarks to the Author):

The premise of this work is to determine the mechanisms underlying alpha-synuclein's ability to interact with mitochondria and drive neuropathologies associated with Parkinson's disease (PD). The study utilizes a human pluripotent-derived A53T-SNCA DA neuronal model versus a genetically corrected cells previously generated by the

authors' to control for potential confounds associated with genetic background. The authors' report that prolonged interaction between alpha-synuclein oligomers and cardiolipin (CL) located on the mitochondrial surface in this model results in lysosomal mitophagy. Furthermore, co-culture of these mutant cells with genetically corrected isogenic neurons results in transmission of mitochondrial pathology to these cells that is blocked by alpha-synuclein antibody suggesting non-cell autonomous transmission of mitochondrial phenotypes through alpha-synuclein to normal neurons. The techniques used to morphologically analyze these cells are all state-of-the art.

The authors state that the link between mitochondrial dysfunction and synucleinophagy remains unclear. However there appear to be several articles which not only link the two, but also explicitly demonstrate a specific interaction between oligomeric alpha-synuclein complexes and the mitochondrial phospholipid cardiolipin (CL). Indeed, in one publication (Zigoneunu et al., 2012), the authors report that the interaction specifically involves the KAKEGVVAAAE repeat region of the N-terminus of the alpha-synuclein protein and acyl cardiolipin side chains. Reported outcomes of this interaction include both increases in mitochondrial fragmentation and APL activation, although the precise mechanisms involved were not determined in these previous studies (Nakamura et al., 2011; Grey et al., 2011; Plotegher et al., 2014, Ghio et al., 2016).

To better explain our results in the context of the existing literature on this subject we have drastically altered our discussion. In our revised experimental data set, we also evaluate competition between synuclein and LC3 with respect to cardiolipin binding. We show that WT α -syn and LC3 compete for binding on cardiolipin, identifying a mechanism by which α -syn may regulate LC3-induced mitophagy. A53T and E46K α -syn had a significantly reduced ability to competitively inhibit LCB binding to cardiolipin, which we believe explains the increase in mitophagic flux observed in *SNCA*-mutant neurons. We now discuss these findings in the context of the excellent work highlighted by the Referee. This discussion can be found on pages 23 and 24 of the revised manuscript, highlighted in blue text.

Additional published data demonstrates that alpha-synuclein oligomers are sequestered by lysosomes and can be transferred from neuron-to-neuron by tunneling nanotubules (TNTs) within these vesicles (Abounit et al., 2016). Other researchers have suggested that cell-to-cell transmission may involve either direct cellular exocytosis-endocytosis of alpha-synuclein protein or delivery via exosomes (e.g. Quek and Hill, 2017; Spencer et al., 2017). All of these studies should be discussed in the context of the authors' own findings.

We have also expanded our discussion to contrast our findings on synuclein transfer with existing work on this subject. Indeed, we have added data to the manuscript showing the capture of free synuclein from conditioned media (Supplementary Table 2), thus suggesting that synuclein is being secreted, at least in part, in a free form. We also include data showing that some synuclein is colocalized with lysosomes in A53T mutant hNs (Supplementary Fig 2), which when considered in conjunction with the finding that mAB-treatment resulted in a partial rescued of synuclein transfer, supports the notion that

there are multiple mechanisms of synuclein seeding that collectively contribute to transmission of pathology. This discussion can be found on pages 25 and 26 of the revised manuscript, highlighted in blue text.

Chu and her colleagues previously demonstrated that under conditions of mitochondrial stress, IMM CL translocates to the OMM ('cardiolipin exposure') and that this in turn lowers the OMM surface charge. CL externalization has also been reported to allow mitochondrial interaction with alpha-synuclein (Nakamura et al., 2011). The authors' demonstrate that in their A53T cell model, CL externalization in response to A53T at early developmental stages coincides with reduced OMM charge. They suggest based on their data that CL externalization is an early stress-induced response to presence of alpha-synuclein which allows its re-folding into a functional alpha-helical state (already suggested by Nakamura's previous work) and prevents subsequent aggregate formation, but that this is inhibited in the presence of A53T mutation which has lower binding affinity for CL. This appears to precede reduced MMP, fragmentation and mitophagy. As Chu has previously shown, CL also acts as an anchor for LC3 and this may explain why A53T results in increased mitophagy and elevated alpha-synuclein aggregation. This is a somewhat novel idea which would be bolstered by inclusion of evidence in OMM LUVs that LC3 binding increases in the presence of increased CL concentrations and that this is inhibited in the presence of WT alpha-synuclein to a greater degree than A53T. One aspect that is still missing in the current study is how alpha-synuclein mechanistically elicits mitochondrial stress to begin with resulting in subsequent CL externalization, since this latter process appears to be required for alpha-synuclein binding to the mitochondria.

As suggested by the Referee and described above, we have expanded our mechanistic investigation with respect to how mutant syn initiates mitochondrial dysfunction. We now show that WT α -syn and LC3 compete for binding to cardiolipin, identifying a mechanism by which α -syn may regulate LC3-induced mitophagy. In addition, we find that A53T and E46K α -syn have a significantly reduced ability to competitively inhibit LCB binding to cardiolipin, which we believe explains the increase in mitophagic flux observed in *SNCA*-mutant neurons. These data can be found in the new figure 5k and l. We further propose that the increased abundance and duration of cardiolipin exposure on the OMM needed to refold mutant α -syn would alter membrane dynamics and initiate the depolarization of mitochondrial membranes and associated mitochondrial stress that we observe in *SNCA*-mutant neurons. This has been added to the discussion of the revised manuscript on page 24.

The authors further demonstrate that co-culturing A53T cells with genetically corrected (Corr) neurons results in increases in detrimental mitochondrial phenotypes, demonstrating transfer of effects of mutant alpha-synuclein from diseased to non-diseased cells. This is blocked by alpha-synuclein Ab suggesting that transmission is via the protein itself (although this is not verified, see below). This is an important concept in terms of whether alpha-synuclein immunotherapy may also prevent mitochondrial defects associated with PD or whether cell-to-cell transmission occurs via other means including nanotubules or exosomes. The authors' findings should be discussed in the context of

previous work showing direct alpha-synuclein protein cell:cell transmission versus other recent studies suggesting occurrence via other means (exosomes, nanotubules).

As outlined above, we have expanded our discussion to contrast our findings on synuclein transfer with existing work on this subject. We have also added data to the manuscript showing the capture of free synuclein from conditioned media (Supplementary Table 2), thus suggesting that synuclein is being secreted, at least in part, in a free form. We also include data showing that some synuclein is colocalized with lysosomes in A53T mutant hNs (Supplementary Fig 2), which when considered in conjunction with the finding that mAB-treatment resulted in a partial rescued of synuclein transfer, supports the notion that there are multiple mechanisms of synuclein seeding that collectively contribute to transmission of pathology. This discussion can be found on pages 25 and 26 of the revised manuscript, highlighted in blue text.

Specific comments:

(1) Note that it is extremely difficult to see and therefore accurately interpret the data panels throughout the manuscript due to their small size. This should be replaced with larger images and include higher magnifications where not presented (some of those that are included are out-of-focus e.g. 1c, 2e, 3f, etc and should be replaces).

We have replaced most of the IF and TEM panels in the manuscript with larger, high resolution panels.

(2) For Fig. 2a,b, sizes of what were designated as fragmented versus unfragmented mitochondria should be included—how were % determined? Also for how % axons containing fragmented mitochondria in Fig. 2e,f.

We have added these details to the methods section on page 27 and to the supplementary methods on page 9.

(3) It is perplexing that no p-129-syn staining and very few fragmented mitochondria are present in the representative panels of control axons in Fig. 2e as a quantitative value of 25% is given for a-syn-129 localized with fragmented mitochondria in cells in 2f. In contrast, Fig. 1 suggests presence of a-129 co-localizing with ubiquitin in the quantitation data graphs of the corrected neurons. Fig. 2g is difficult to see and requires some type of quantitation.

A higher resolution micrograph of panel 2g has been inserted which shows the levels of PS129 synuclein in corrected cell more clearly.

(4) It does not appear that soluble oligomers were present in mitochondrial fractions from either A53T or Corr cells (Fig. 3b). This appears in contrast to what has been previously published. It would be helpful if the entire gel was included here.

We saw no evidence of multimeric, higher molecular weight synuclein forms in Tx100 soluble lysates from the mitochondrial fractions, in either corrected or A53T samples. We did see evidence of a 60 kDa synuclein that may represent the tetrameric conformation of synuclein that has been reported by Bartels et al. (2011) Nature 477(7362):107-110. However, the absence of multimeric bands makes us reluctant to describe these as soluble oligomers. In addition, we see an abundance of soluble 14 kDa synuclein in the mitochondrial fractions. We cannot rule out that some soluble oligomers were present in mitochondrial fractions and that Tx-100 solubilization disrupted their structure, and as a result, that the 14 kDa band we see represents a combination of monomeric and multimeric syn. As a result, we have removed reference to monomers from this section of the text, as we can only say with certainty that the synuclein is in the soluble form. The full blot is below for the Referee's consideration.

(5) What happens when labeled A53T cells are co-cultures with A53T cells? Are elevated levels of alpha-synuclein inclusions and mitochondrial phenotypes observed versus in Corr cells co-cultured with A53T?

The data in figure 2a can be considered representative of this experiment. For the experiments in figure 2a, cells were infected with MitoDSRed expressing lentivirus at the NPC stage, at low titer, to permit imaging of the mitochondria from individual neurons upon terminal differentiation. As such, this is not unlike differentiating labeled A53T cells with unlabeled A53T cells in co-culture. On average, 80% of A53T neurons were fragmented under these culture conditions. While A53T cells stably selected and co-

cultured with corrected cells (Figure 6d) had only 65% fragmented neurons, the difference between these values was not statistically significant given the degree of variation between coverslips.

(6) Levels of alpha-synuclein in the conditioned media from A53T versus Corr cells should be quantified +/- Ab to verify that observed non-cell homologous effects are truly due to its secretion from the former.

These data are now presented in supplementary table 2.

REVIEWERS' COMMENTS:

Reviewer #2 (Remarks to the Author):

The level of additional data for resubmission is extensive. I am impressed by the additional of the extra lines, mutations and isogenic controls. The authors have therefore answered my major concerns.